# Improving Security of Future Networks Using Enhanced Customer Edge Switching and Risk-Based Analysis

Slawomir Nowaczewski [ID] and Wojciech Mazurczyk *[ID]

Institute of Computer Science, Warsaw University of Technology, Nowowiejska 15/19, 00-665 Warsaw, Poland; slawomir.nowaczewski@pw.edu.pl
* Correspondence: wojciech.mazurczyk@pw.edu.pl

**Abstract:** Customer Edge Switching (CES) is an extension of the already known classical firewall that is often described and used in future networks like 5G. It extends its functionality by enabling information exchange with other firewalls to decide whether the inspected network traffic should be considered malicious or legitimate. In this paper, we show how the Passive DNS can be used to further improve security of this solution. First, we discuss CES solution and its internals. We also determine how it uses DNS and CETP protocols. Secondly, we describe the basics of the Passive DNS and how it impacts the DNS protocol. Thirdly, we evaluate how the Passive DNS can be extended to collect also CETP information. Finally, we integrate the solutions and present obtained experimental results.

**Keywords:** Custom Edge Switching (CES); passive DNS; DNS; passive CETP; CETP; risk-based; analysis engine; 5G; future internet



## 1. Introduction

Nowadays, it is evident that 5G is the next step in the mobile networks' development. It is also already well-known that with this step new security problems will arise. should provide us then with much better security than it is observed in the current mobile networks [1,2]. Therefore, new countermeasures have to be developed that would be able to address new and emerging threats. One of such countermeasure is Customer Edge Switching (CES) which extends the classical firewall functionalities by using policy-based communication with other firewalls to guarantee the requirements of both sides are met [3,4]. Note, that currently in communication networks, e.g., in the Internet, the sender can transmit practically anything to the receiver with only limited inspection depending on the assumed security policy. However, CES requires that both sides of the communication negotiate all the parameters first before appropriate data exchange is performed. Two firewalls, that are involved in the communication process, negotiate and decide which traffic is benign or malicious. This negotiation functionality is crucial and can defend from attacks such as Denial of Service (DoS) or address spoofing [5,6]. The architecture of the CES is quite straightforward. It extends the functionality of the network edge currently occupied by classical firewalls and it can operate in various configurations [5,6]. Note, that for CES two options are possible, i.e., that both sides of communication are utilizing CES or only one side is using it. Hence, we can discuss CES-to-CES, CES as NAT [7], or CES as Realm Gateway (RGW) scenarios. The architecture of the CES is illustrated in Figure 1 where two CES firewalls that separate customers private networks from the provider network (CES-to-CES communication scenario) can be observed.

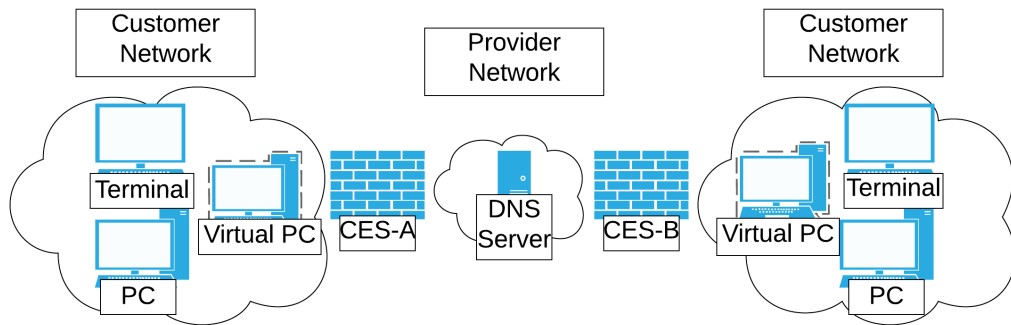

**Figure 1.** Customer Edge Switching Architecture [8].

As it was already mentioned, CES can be treated as an extension of the classical or the converged firewall [9,10]. Hence, it also supports NAT functionality. CES, in order to enable communication, translates private to public addresses. It creates flow states for each seen initial packet that is being allowed to go through the node. Moreover, if it notices the returning packet for the flow already present in the flow table it just forwards it. For the situations where no record exists in the flow table, no packet is allowed to pass through.

Customer Edge Switching is policy-based solution. This means that before any data exchange takes place, the parameters of the communication must be negotiated. The optimal case is when both sides of the data exchange use CES functionality. If only one of them has implemented CES, it is a legacy or transitional functionality depending on where CES is located. All possible configurations are presented in Figure 2.

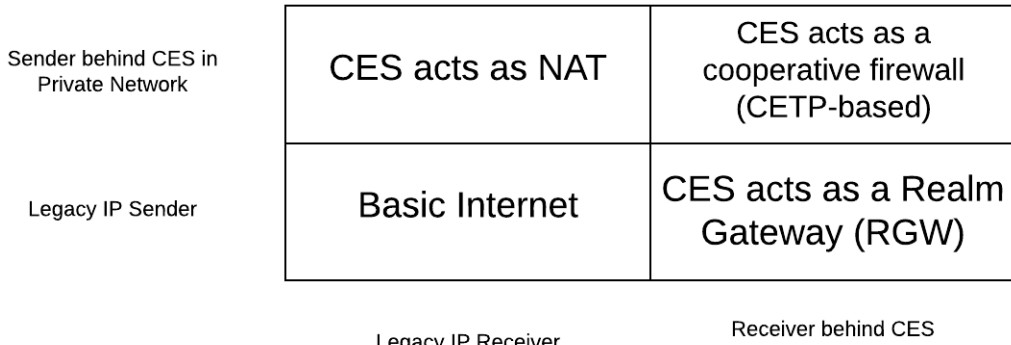

**Figure 2.** CES and NAT comparison [8].

CES can perform some additional operations in order to allow or to block packets. It is said that these are the extensions of the traditional firewalls [11,12]. CES can check if the policy compliance is met and if the packet is already authenticated or is it being spoofed. This is what negotiation enables. CES exchanges special queries and determines if the host policy requirements such as *Id.fqdn*, *ctrl.cesid* or *rloc.ipv4* match. These queries can also be sent to other nodes such as Active Directory (AD) or Certificate Authority (CA) [13]. There are many ways to build policies within CES. They can be static or dynamic. They can be also enabled at the CES- or at the host-level. Depending on how the policy is built, CES can act differently for each packet it inspects [14].

CES serves also as a proxy node which separates hosts and applications on the internal network from these placed on the external network. Proxy functionality is mainly implemented using Customer Edge Traversal Protocol (CETP) in order to exchange policies before any communication with a host or an application takes place. CETP exchange must occur before every data exchange. CETP can also exchange queries and inform other CES nodes about malicious traffic. Hence, it is able to block the transmission as close to the source of traffic as it is possible which is desired as otherwise a significant volume of unwanted traffic may be transmitted within the defended network [15].

Considering above, the main novel contributions of this paper are as follows:

1.  Passive DNS with CETP support—we propose and describe how Passive DNS can be enriched with inspection of the CETP traffic. Combining Passive DNS with CETP support is an innovative solution.
2.  Risk-based Analysis Engine for Passive DNS/CETP—we introduce a new mechanism for risk-based study and benefits of the risk-based detection. To the best of the authors' knowledge, there is no other similar solution.

The rest of the paper is structured as follows. Section 2 outlines the most relevant works related to the research presented in this paper. Next, in Section 3.6 security analysis of CES is presented, while Section 3 showcases briefly the most important information on CES and related aspects. Then in Section 4 the proposed solution of how to apply Passive DNS to CES for improved security is presented. The concept of risk-based analysis engine for the Passive DNS/CETP is illustrated in Section 5. Section 6 outlines experimental testbed, implementation, and methodology, while Section 7 presents obtained results. We conclude this part with presenting the relevant case study in Section 8. Finally, Section 9 concludes our work and points out potential future research directions.

## 2. Related Work

In this section we describe the most relevant, existing works that are closely related to the research presented in this paper. First, we analyze various network architectures for next-generation networks and then we focus on outlining different previously proposed anomaly detection approaches.

### 2.1. Network Architectures

Customer Edge Switching is the receiver-oriented solution. As already mentioned, it means that before any data exchange takes place, both sides of the communication must negotiate appropriate parameters. This is also known as the publisher-subscriber architecture as nothing is being delivered if the policies were not exchanged [16]. Obviously, CES is only one of the possible architectures to ensure the communication. There are many others described in previously published papers. For instance, Metis is based on Software Defined Network (SDN) and is a solution straight from the 5G-PPP [17]. Locator/Identifier Separation Protocol (LISP) is relatively known architecture which separates Endpoint Identifiers (EID) from the Routing Locators (RLOC) [18]. Similarly to CES, LISP also is based on DNS. Next architecture, IPNL, is an extension to NAT and ensures communication by combining public IP addresses with domain names [19]. Another approach, i.e., I3 uses rendezvous server to enable the communication by spanning nodes between sender and receiver [20]. Shim6 and HIP, respectively, add extra locator below the transport protocol [21] or use identity tags [22]. There are also two other architectures: TRIAD which is based on source routing and names [23] and MILSA which uses functional roles linked to connectivity domains [24]. StopIt is based on filters implemented on the edge of the network [25]. These filters can be uploaded to the sending node to block traffic as close to the source as possible. Last two solutions, SIFF and PBS, prioritize tagged packets [26] or proactively filters packets based on NSIS protocol suite for signaling [27,28], respectively. The state-of-the-art comparison between different approaches is shown in Table 1 [8].

**Table 1.** State-of-the-art comparison between the approach and key solutions for the different network architectures [8].

| Parameters/ Approaches | Author | Decoupling Location and Identification | Additional Protocol Layer | Architecture | Key Tasks |
|---|---|---|---|---|---|
| IPNL [19] | Francis and Gummadi (2001) | Globally routable IP addresses with domain names | Yes | NAT-extended | Revealing private IP addresses |
| I3 [20] | Stoica et al. (2004) | Yes, Decoupling senders from receivers | Yes | Overlay network with a rendezvous server | Mobility, multicast and anycast |
| TRIAD [23] | Gritter and Cheriton (2001) | Yes, Name with source routing | Yes | Address realms | - |
| MILSA [24] | Pan et al. (2008) | Yes, ID with Locator | YesNew architecture | Functional roles for trust domains and connectivity domains | - |
| LISP [18] | Farinacci et al. (2013) | Yes, Endpoint Identifiers (EID) with routing locators (RLOC) | Yes, LISP header | Endpoints and routing locators | Encapsulation, rewriting and mapping of IP addresses |
| Shim6 [21] | Nordmark and Bagnulo (2001) | Yes, Additional locator | Yes | Load sharing and multi-homing | - |
| HIP [22] | Moskowitz and Nikander (2006) | Yes, Additional locator | Yes, HIP header | Cryptography and identity tags | Authentication and protection from DoS |
| StopIt [25] | Liu, Yang and Lu (2008) | No | Yes | Filtering at the edges | Blocking unwanted traffic |
| SIFF [26] | Yaar et al. (2004) | No | Yes | Tagging and prioritizing packets | Protection from DoS |
| PBS [28] | Hong and Schulzrinne (2013) | No | Yes | Filtering and control plane based on NSIS protocol | Heavy cryptography |
| Metis [17] | Osseiran and Boccardi (2014) | No | Yes | Control plane based on SDN | Software-defined network control |
| CES with DNSCrypt and DNSSEC [8] | Nowaczewski and Mazurczyk (2020) | Yes | Yes | CETP protocol | - |

## 2.2. Anomaly Detection

Anomaly detection research began with statistical research [29–31]. The concept of anomaly detection in the computing environment started a little later [32–34]. Work on the detection of anomalies raised the issues of advantages and disadvantages of various methods or the comparison and evaluation of various techniques [35]. Other papers, in turn, penetrated deeply into individual techniques and applications [36]. Research presented in [37,38] focus on the use of Machine Learning (ML) techniques in the detection of anomalies or detection in the IoT environment [39].

## 3. Technical Background

The identification of hosts and applications in CES is being performed using Fully Qualified Domain Names (FQDN). FQDN implies utilization of the DNS protocol. CES stores information about hosts and applications in the network it supports thus, it can respond to the DNS queries it receives. A DNS query is the basis for the CETP Service Discovery which is a functionality of the CES that is executed before any policy negotiation. This function verifies the remote network and if the appropriate hosts/applications can be reached. CETP + cisid format is used to make every available service visible. CETP uses Naming Authority Pointer (NAPTR), a field in the DNS packet format, to communicate with the service on the other side [11]. NAPTR response holds the IP address and other fields. The public IP address is called Routing Locator (RLOC) and it is used to reach the remote CES. RLOC for IPv4 has "ip=100.100.100.100" format but it can also take other forms such as ports or IPv6. NAPTR can also transport alias field. CES can keep the DNS records within the node but the records can also be located on the separate DNS proxies or servers [40].

### 3.1. CETP Communication

CETP is responsible for signaling and transmission in the CES-to-CES scenario [41]. The first function, signaling, is needed to negotiate parameters before any data exchange. In this scheme, two CES nodes must agree to the defined policies. The second function, transmission, allows encapsulation of data packets within CETP and exchanging them between two nodes. CES uses different session identifiers to process all the sessions it handles [42,43].

### 3.2. CETP Packet Format

CETP packet structure begins from 4-byte header that includes the protocol version, two flags and the header length. Then we have the Source Session Tag (SST) and the Destination Session Tag (DST) which are used to uniquely mark session directions. After these fields there is either the data payload or signaling information. Data payload and signaling information are encapsulated within the Type-Length-Value (TLV) structures. These structures can also contain appropriate policies. Each TLV is grouped in the Type class. Further it can be divided into operation, group, and code fields. The operation field can be set only to info, query or response values. These values are used when the policy needs to be announced (info), requested (query) or just sent in response (response). The structures of the CETP payload TLV and the CETP control plane are illustrated in Figures 3 and 4, respectively. Policies elements that can be transported are listed in Table 2.

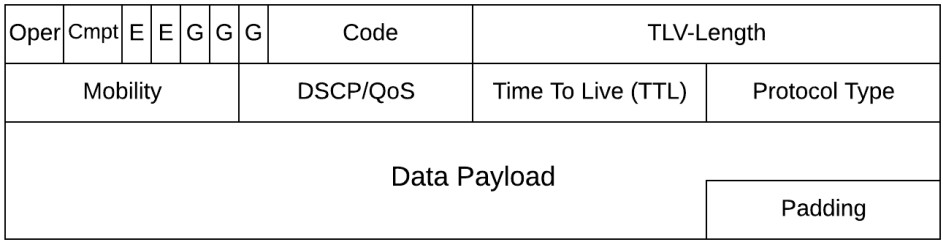

**Figure 3.** CETP Payload TLV [8].

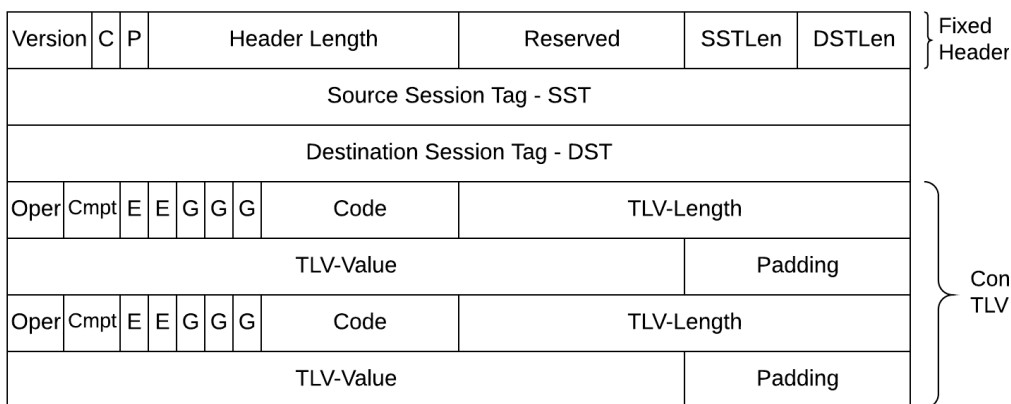

**Figure 4.** CETP control plane structure [8].

**Table 2.** CETP policy elements [8].

| Group | Code | Description |
|---|---|---|
| CES | - Pow <br> - cesid <br> - headersignature <br> - cace | - The proof-of-work computation <br> - FQDN-based ID of the CES node <br> - Signature of the CETP packet <br> - The CA address for CES validation |
| Control | - Dstep <br> - caep <br> - terminate <br> - warning <br> - ack <br> - ttl <br> - ratelimit | - FQDN-based destination endpoint ID <br> - The CA address for endpoint validation <br> - Contains session terminating information <br> - Contains the warning information <br> - The acknowledgement number <br> The time to live for session <br> - The rate limit for session |
| CID | - Fqdn <br> - maid <br> - moc <br> - msisdn | - FQDN-based ID of the sender <br> - The Mobile Assured ID <br> - The Mobile Operator Certificaten <br> - The MSISDN number of the host |
| RLOC | - IPv4 <br> - IPv6 <br> - eth | - An IPv4 address (RLOC) of the CES <br> - An IPv6 address (RLOC) of the CES <br> - An MAC address (RLOC) of the CES |
| Payload | - IPv4 <br> - IPv6 <br> - eth | - IPv4 encapsulation of the user payload <br> - IPv6 encapsulation of the user payload <br> - Eth encapsulation of the user payload |

*3.3. Policy Negotiation*

In Figure 5 it is visible how the whole negotiation process in the CES-to-CES scenario is performed. There are three main phases during which CES establishes the communication and one phase for data exchange. First, the discovery of the CES endpoint is started when the DNS query with A record from the supported host is sent. The CES evaluates this packet and replaces the A record with NAPTR record which is used to signal the availability of the services. In NAPTR record we can find, for instance, such information as a service (CETP+cesid), CES identifier, endpoint, or alias. The DNS response contains RLOC address that is used to reach the other side. Secondly, after discovering CES endpoint, the negotiation phase is started which is just a mutual exchange of policies between two CES nodes. This is where the CETP signaling is performed. The CETP control plane is organized into the structure visible in Figure 4. What is important, the policy-based communication in CETP requires three functional substructures, or simply saying vectors, for policies: *Offer*, *Require*, and *Available*. These substructures are used depending on the direction where the CETP packet is travelling to. Each of these vectors contain policy elements that have been

shown in Table 2. The exchange also allocates IP proxy addresses for the data exchange if the negotiation is successful or inform about the error in the NXDOMAIN field of the DNS response packet if it is not. There are also some additional fields in the header of the CETP Payload TLV such as Mobility or DSCP/QoS. In general, during negotiation process, the initiating CES firewall (CES-A) sends three different vectors of policies to the receiving side: *offer*, *requirements* and *available* policies. It also sets Source Session Tag (SST). The responding CES (CES-B) sets Destination Session Tag (DST) and sends appropriate policies in the required and available groups. Thirdly, the initiating CES sends correct proxy address that the sender can use to communicate with the destination. Finally, both sides of the communication can transfer data. Data between two CES nodes is exchanged in CETP protocol, however, note that it is transparent for both hosts. Host-to-host policy negotiation occurs after negotiating the policies by CES nodes. Policies are a good solution to the problems of DoS attacks, misbehaving hosts, and address spoofing. They are also the foundation of the CES. There are many different kinds of policies that can be configured and used. For example, we can define the CES-level main policy but we can also focus on more specific host-level, application-level, or service-level ones [44–46].

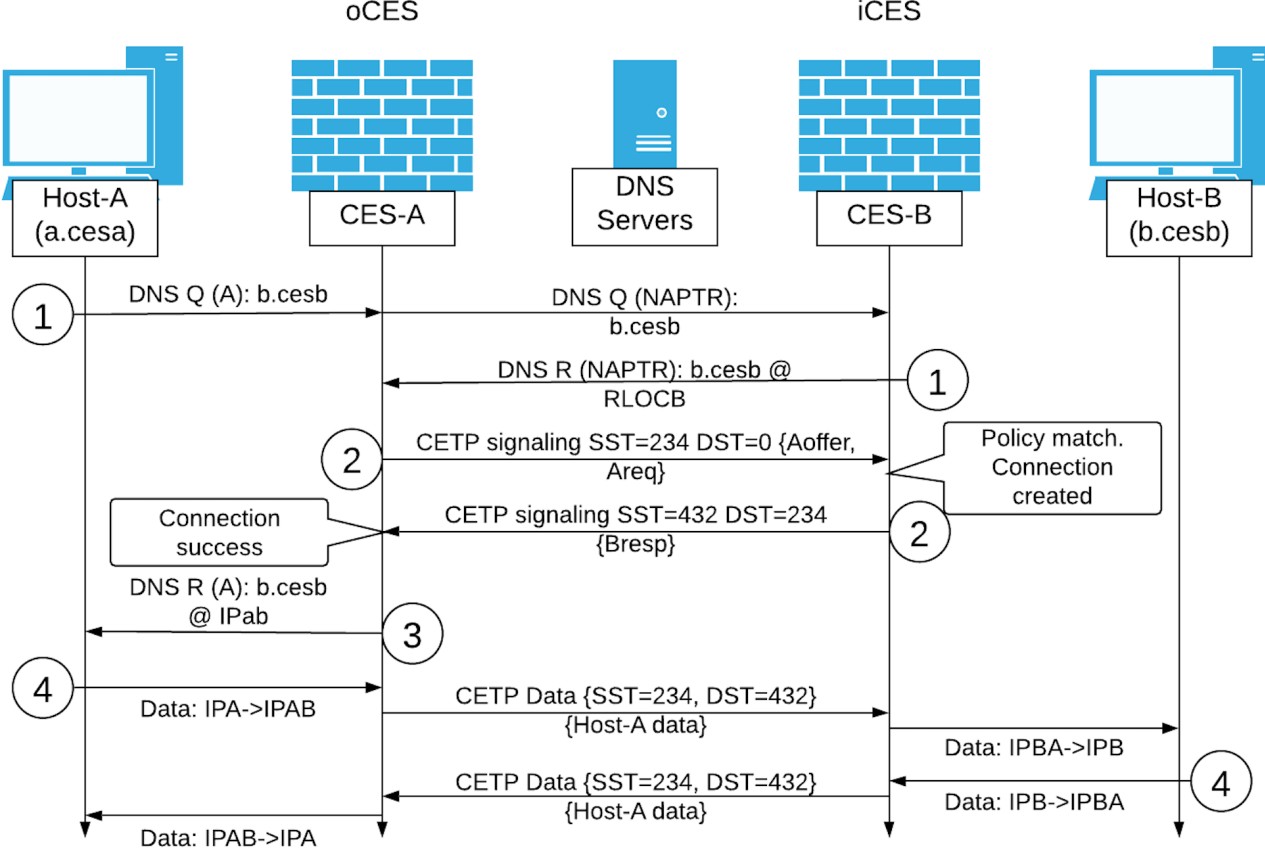

**Figure 5.** CETP signaling–scenario with CES-to-CES communication [8].

### 3.4. Security Mechanisms in CES

Customer Edge Solution is a policy-based solution. It negotiates all policy elements on behalf of hosts. The negotiation of policies guarantees that no data is being exchanged before successful initialization of the communication. Hence, address spoofing and DoS threats are easily eliminated. Additionally, CES provides to administrators a lot of tools in one place that are helpful in configuring appropriate policies and maintaining security. First of all, CES collects many logs. These logs are helpful in building reputation and trust of the senders. Reputation is desirable information to provide better security of private networks.

CES contains a lot of different security mechanisms that can help mitigate the risk of successful attacks [47]. These countermeasures can help with elimination of the spoofed or unwanted traffic [48,49]. They are policy-controlled and are helpful when checking routing using secure locators or ID types. A few of these security mechanisms have been mentioned in the list below:

1.  Policy-based Communication: this mechanism ensures that before any data is exchanged a negotiation should take place. It prevents traffic from unauthenticated hosts or unwanted sources.
2.  CES Authentication: a solution that requires authentication of the local CES to the remote one. This can be achieved with the use of CA and X.509 certificates.
3.  Header Signature: the signature provides the integrity of the files. In this case, the receiving CES can check if the packet was modified while travelling communication path.
4.  Proof-of-Work: a mechanism that pushes most computing to the sender instead the receiver as it is observed in the current Internet.

With respect to CES vulnerabilities, we can divide them roughly into 2 groups: (i) CES-based and (ii) Legacy Host attacks. The difference between them is in Virtual Private Network (VPN) and if it is shared with CES devices or not. If the VPN is shared, legacy hosts can generate CETP attacks. Legacy host which is present in the middle of the communication path can modify the content of CETP packets. The packets can be also replayed. Both of these attacks can be neutralized using unique cookie mechanism to authenticate the correct sender. Additionally, legacy host can present itself as CES and communicate with the other side. CES verification can prevent such attacks as it was already mentioned in the list. The Man-in-the-Middle (MitM) attack is a serious threat and many its variants exist, e.g., via BGP poisoning, DNS cache poisoning, etc. The countermeasure is to use digital signatures in the packets, certificates, and Public Key Infrastructure as it was already mentioned. As CES has many functionalities, there is also a class of attacks that exhausts the resources of the node. Circular Pool of Public Addresses (CPPA) algorithm which enables initiation of inbound connections towards internal hosts and which can be triggered by incoming DNS queries can be exhausted by sending many requests.

In this paper, we discuss Passive DNS functionality and how it can be extended to collect information from CETP. We show how this solution can be integrated with CES architecture for improved security. Thanks to the Passive DNS/CETP module the whole investigation process takes definitely much less time. The module helps to discover attacks using DNS and CETP protocols and enables to easily identify relevant communication sessions in order to evaluate whether unwanted data were included in CETP packets. It also allows to analyze statistical data related to DNS/CETP sessions and in case of any anomalies to undertake rapid response actions.

*3.5. Realm Gateway*

Realm Gateway (RGW) is one of the possibilities how CES can be used in practise. It is used as an intermediary step to implement the complete CES-to-CES solution which is the most recommended variant. In this scenario, CES is located only on one of the communication sides. It behaves as usual NAT device for outbound traffic but for inbound traffic it uses Circular Pool of Public Addresses (CPPA) and provides better security [50–52]. CPPA algorithm starts when the CES receives the DNS query from the outside of network it is securing [53]. CPPA algorithm can support three types of incoming traffic:

1.  a usual DNS query packet with FQDN of the host or service,
2.  HTTP(S) traffic which can be served by the reverse HTTP proxy,
3.  a mapping similar to port the forwarding or to the port overloading in NAT.

For each received DNS query, CPPA allocates a public IP address. Then, CES can respond with the DNS response packet which contains this IP address and during this

process it also creates internally a state for an incoming connection. This state is a half-connection state. CES keeps this state until data packets come. If the CES notices such packets, the state is changed to full-connection state and the traffic can be exchanged between private and public hosts without any issues [54]. The complete process of handling the connections in the RGW scenario is illustrated in Figure 6. Both 1 RTT and 2 RTT scenarios are visible in Figures 7 and 8, respectively.

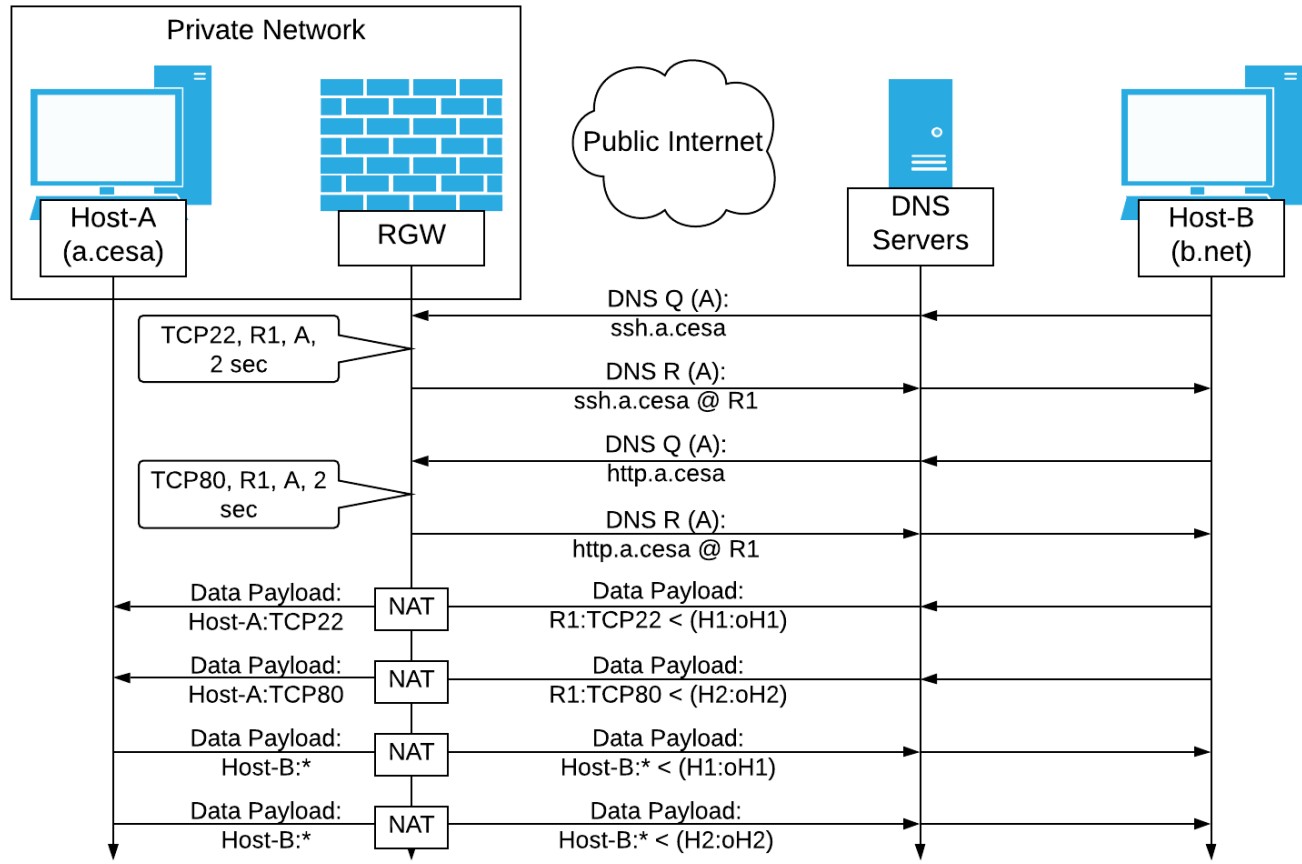

**Figure 6.** CETP signaling–scenario with RGW (oCES-Outbound CES, iCES-Inbound CES.

*3.6. Security Mechanisms in RGW*

RGW operation may be disrupted by using the DNS protocol [44]. The most obvious attack is DNS flooding or just DoS with source address spoofing. When the CES receives a plenty of DNS queries, CPPA algorithm must allocate an IP address for each query it receives. The flooding of the DNS queries can lead to quick resources consumption [55]. Hence, new connections coming from legitimate sources will be blocked [53]. RGW has a functionality that can detect such attacks. It uses internal states table which contains information about every connection. If the RGW receives a packet for which it cannot build or find an entry in the table, it will just block it or blacklist it depends on the configuration. RGW uses a reputation mechanism to rate the servers and three types of lists that support this: whitelist, greylist, and blacklist. It also depends on the configuration of these lists how the traffic from different servers/hosts is served. Whitelisted traffic is served, blacklists is blocked, and greylist is a list that includes everything that uses appropriate policies.

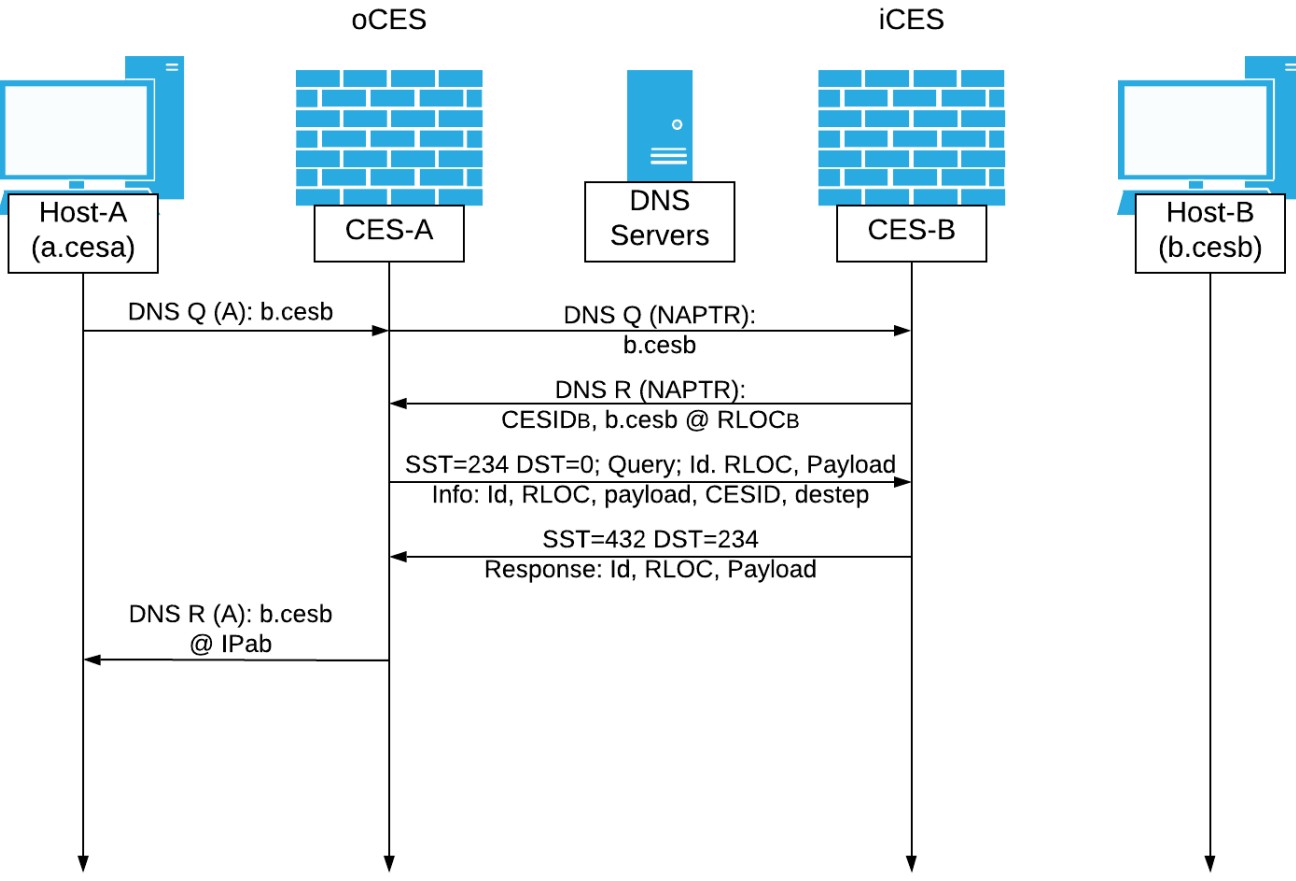

**Figure 7.** CETP connection establishment—1 RTT [8].

Note, that there are also a few other security mechanisms in the CES. The first, known as the TCP-splice method, is a mechanism for checking or challenging the sender by sending cookie embedded in the Initial Sequence Number (ISN). Another method is rate-limiting CPPA which can prevent DoS attacks. CES also contains a mechanism that prevents connection hijacking if it detects anomalies in the received packets after comparing them with the states table. The last mechanism described in this work prevents creation of entries in the states table if the host or the server were not authenticated first. This solution can also block malicious nodes.

*3.7. Passive DNS*

Passive DNS is a technology that allows to observe captured DNS queries and based on them build relationships with the DNS responses in order to collect, store, and present valuable information. It provides better visibility into what is really happening in the network. This visibility allows to quickly evaluate what IP addresses were assigned to the particular domain name, which more specific FQDN were observed under the base FQDN or what domain names were received from the particular authoritative server. Passive DNS does this in the form of reports with created relationships. Passive DNS is often used to enhance threat intelligence (the knowledge about existing and emerging threats; especially their mechanisms, context, indicators, and the advises how the threats can be mitigated). It collects information from the expected and unexpected DNS packets, links them respectively and allows to filter the results as it is required by the administrator. Then it is possible for the administrator to link information from the Passive DNS with others sources such as abusing IP addresses, malicious domain names, or name servers. Passive DNS is using a database to store the collected information. This means that if the

administrator wants to create a report he may analyze historical data even for a long time periods. This information can be indispensable when looking for malicious activity in the network especially that all DNS fields can be stored. It can be stated that Passive DNS provides better information than reputation systems as it shows what was really seen in the network, not a generated reputation data that can sometimes reflect incorrect view. The reputation data about domain names and IP addresses can always be added to the information from the Passive DNS during enrichment process. Sample outputs from the Passive DNS are visible in Tables 3 and 4. The relationship between the DNS query and the DNS response can be clearly observed there. Additionally, there are some other added valuable metadata like First Seen or Last Seen. Of course, the data or the metadata that needs to be collected can be appropriately set – there are many possible options. It depends on the business requirements what kind of data and metadata need to be stored in the database and how they should be related to the other fields. There is a possibility to collect every DNS protocol field and they can be queried from the database in various ways using, e.g., SQL queries.

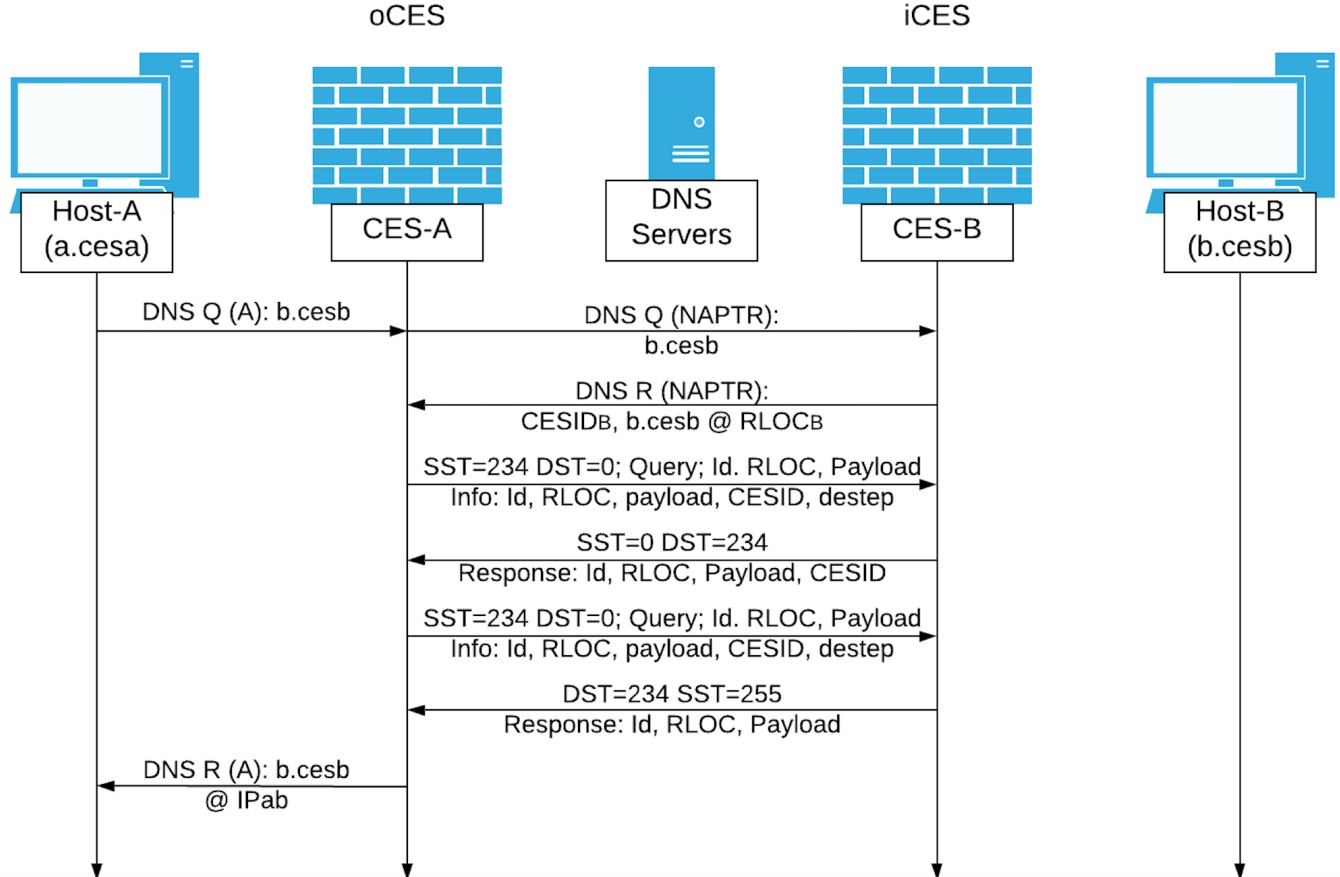

**Figure 8.** CETP connection establishment—2 RTT [8].

**Table 3.** Exemplary output of the Passive DNS. First view.

| Timestamp | DNS-Client | DNS-Server | RR class | Query |
|---|---|---|---|---|
| 1,588,322,540 | 192.168.0.1 | 8.8.8.8 | IN | google.com |
| **Query Type** | **Answer** | **TTL** | **Count** | |
| A | 172.217.20.174 | 57 | 14 | |

**Table 4.** Exemplary output of the Passive DNS. Second view.

| First Seen | Last Seen | Query Type | TTL | Query |
|---|---|---|---|---|
| 1,588,322,540 | 1,588,322,555 | A | 60 | google.com |
| **Answer** | | | | |
| 172.217.20.174 | | | | |

*3.8. Challenges in Anomaly Detection*

Several major challenges are related with network anomaly detection. Some of them are discussed below. The Masking effect and Swamping effect, one of the most important effects that take part in anomaly detection and affect the amount of false positives and false negatives, are discussed in more detail below and they are shown in Figure 9.

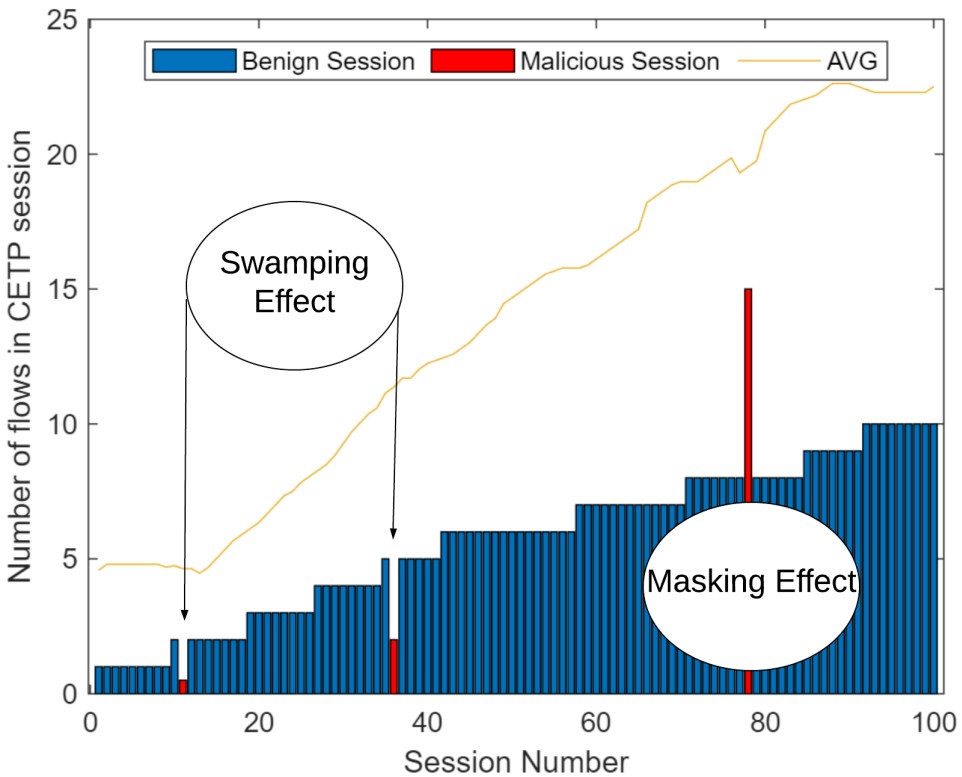

**Figure 9.** Masking and swamping effects.

3.8.1. Masking Effect

One network anomaly masks a second anomaly where the second anomaly can only be treated as an anomaly without the former being present. An example would be when an attacker introduces carefully crafted flows into the network that artificially raise the threshold level. The threshold is adjusted to the artificial flows, so that its threshold is higher than the true anomaly and in result the anomaly cannot be detected [56].

3.8.2. Swamping Effect

A situation that is opposite to the masking effect. It consists in the fact that anomalies take values very low in relation to their neighbors or, in other words, are immersed in relation to their neighbors. An example of this type may be a situation similar to the masking effect, when the attacker artificially introduces flows with higher values in a specific domain, which ignores anomalies with low values [56].

### 3.8.3. Frequency of Anomalies Effect

Network anomaly detection usually runs correctly when the number of anomalies is relatively low in relation to all tested flows. Anomaly detection may not work when in a situation where there is a lot of traffic in the network, which leads to a large masking effect or swamping effect. Such a situation may be implied by, for example, a DDoS attack [56].

### 3.8.4. Lag Effect

As there is a slight delay between the observation of an anomaly and its detection, it may happen that a given anomaly is detected multiple times. This is especially true in very complex systems that operate on many domains. Depending on how many domains we perceive a given anomaly, there may be many delay values, which depend on the number of domains we operate on. Those with fewer domains will be detected faster.

### 3.8.5. Dimensionality Effect

Another effect is related to dimensionality. It consists in the fact that when we increase the number of domains in which we look for anomalies, the data space also increases. This can lead to a situation where malicious flows have been scored low values of risk in the individual domains and appear at the very end of the list. The priority list that contains ordered flows from the highest risk to the lowest will be discussed in more detail in the Section 5.

### *3.9. Risk Scoring*

Risk scoring is a function

$$F_R : X \mapsto R+; \quad X \; is \; tested \; flow \quad \text{and} \quad R+ \; is \; risk \; score$$

that assigns a risk value to each of the flows tested. Based on the risk scoring function, we can prepare a priority list that contains ordered flows from the highest risk to the lowest. The priority list will be discussed in more detail later in the article in Section 5. Some of the possible functions will be discussed in this subsection.

### 3.9.1. Binary Scoring

The simplest possible function. It assigns the value 1 for detection on a particular domain, or 0 when there is no detection. Values from individual tested domains are accumulated, which determines the total achieved risk score.

### 3.9.2. Unified Scoring

Each flow is assigned a value in the range [0, 1] depending on how much the given flow exceeds the threshold. The flow with the highest value in the analyzed domain gains the risk score value equal to 1, and the flow below the threshold-value 0. The rest of the flows, those above the threshold, gain risk score values proportional to the flow with the maximum value obtained. Values from individual tested domains are accumulated, which determines the total achieved risk score.

### 3.9.3. Q-Function Scoring

Assuming the error vector distribution is Gaussian, we can use this to analyze the error vector. First, we predict $m$ values for $d$ dimensions. In the next step, we shift our predictor by observing {t-m, ..., t-1}, which allows us to obtain the values predicted for each point in time. Based on these values, we get the error vector, which is the difference between the obtained value and the predicted value. The obtained error vector

$$(N = N(\mu, \sigma)); \quad N \; is \; Gaussian \; distribution, \; \mu \; is \; the \; mean \; and \; \sigma \; is \; the \; standard \; deviation$$

is compared with the Gaussian distribution on the basis that the probability p(t) of obtaining a given error is equal to the value of N at time t. Anomaly detection is obtained by deciding

whether p(t) is less than T, where T is computed based on the unobserved validation set [57]. We can calculate additional risk score for particular flows based on the received results from Gaussian distribution. Values from individual tested domains are accumulated, which determines the total achieved risk score.

### 3.9.4. Similarity Scoring

This approach assigns the risk score based on the distance of the analyzed flows to groups such as k neighbors. The whole process is to find a similarity in order to distinguish near flows from those far away. Distance can be studied on the basis of Manhattan distance, weighted Euclidean distance, distance in terms of distribution or correlation [57].

### 3.10. Cell Averaging-Constant False Alarm Rate

The data analysis engine created for the Passive DNS/CETP module determines the decision thresholds adaptively using the Cell Averaging - Constant False Alarm Rate (CA-CFAR) algorithm [58]. This algorithm is very good for detecting signals in radars and at the same time it is suitable for the detection in varying network conditions [58]. The detection is straightforward. The algorithm just compares the signal to a threshold. Hence, it all comes down to the determination of the appropriate decision threshold, which is both a function of correct detection and the incorrect detection. In the CA-CFAR detector, noise samples are extracted from two groups called training cells which are located on either side of the Cell Under Test (CUT). CA-CFAR averages the results for individual groups of training cells in estimating noise power. The number of training cells on both sides of the CUT is the same. In addition, the CA-CFAR algorithm uses so-called guard cells. Their task is to prevent the signal from penetrating the training cells, which could adversely affect detection. The relationship between individual cells and the scheme of the algorithm's operation is presented in Figure 10.

The way in which the threshold is determined can be dynamically controlled as it uses the parameters which describe numbers of training cells and guard cells. These can be selected dynamically depending on the current network conditions, e.g., whether there are many ongoing sessions or not.

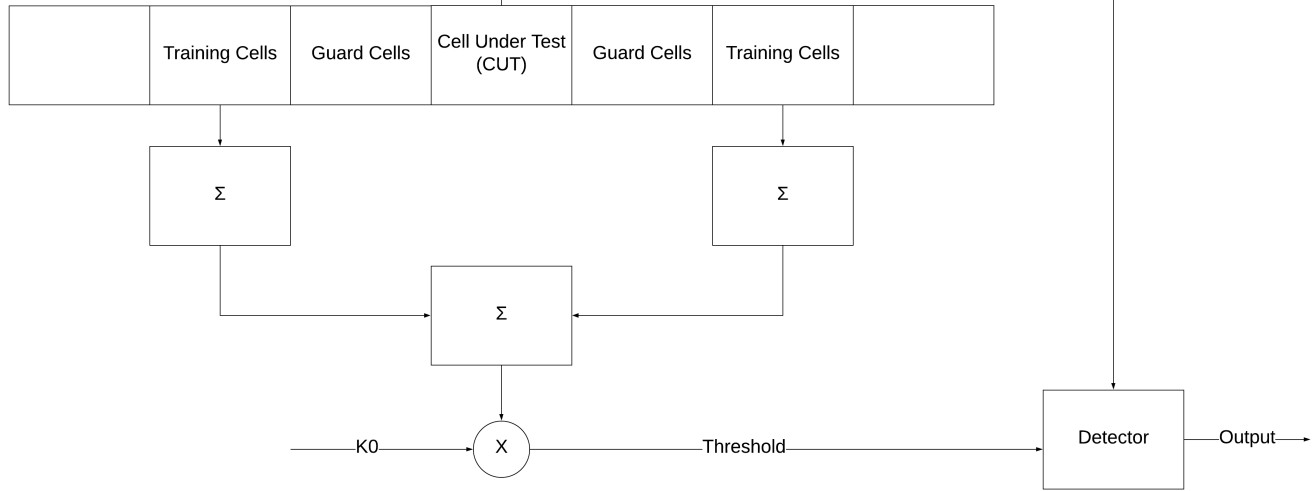

**Figure 10.** CA-CFAR algorithm.

## 4. Passive DNS Support for CETP

As it was mentioned in the previous subsection, the Passive DNS is a technology that allows to observe captured DNS queries and based on them build relationships with the DNS responses. By default, it does not collect information from other protocols. However, in case of CETP, we can modify it and extend the solution to support this protocol. Passive

DNS/CETP can collect many desired information from different phases of CETP signaling and data exchange. First, the discovery phase provides information about A and NAPTR records from the DNS query which is used to check for availability of the services. NAPTR record includes, for instance, information about the service (CETP+cesid), CES identifier, endpoint, or alias. The DNS response contains RLOC address used as the locator of the service. All these additional fields in DNS can be added to Passive DNS/CETP in order to have better visibility what is really happening in the network and how everything is related to each other. Secondly, the negotiation phase provides information about Source Session Tag (SST) and Destination Session Tag (DST). This information, if collected, can help the administrator to quickly identify the appropriate session and the desired traffic. The data that can be carved out by Passive DNS/CETP module contain also IP proxy addresses, the information whether the negotiation was successful or the error from the NXDOMAIN field. Passive DNS/CETP solution can collect information about substructures for policies being used: *Offer*, *Require* and *Available*. The module collects data about operation fields which can be set to *Info*, *Query* and *Response* values. There are also some additional fields in the header of the CETP Payload TLV such as *Mobility* or *DSCP/QoS* which can be useful as well. Thirdly, Passive DNS/CETP can collect information what response was finally sent to the initiating host. Fourthly, Passive DNS/CETP can accumulate metadata and statistical information about the data being exchanged in the session. The solution can collect information properly in 1 RTT and 2 RTT scenarios of the connection establishment which were described in more detail in Section 3.5. Both 1 RTT and 2 RTT scenarios are visible in Figures 7 and 8, respectively. Furthermore, the Passive DNS/CETP gathers historical data from different CETP sessions. These historical data can be extremely useful in the investigation process. If we add a data analysis engine to it, that is included in the module, we can obtain additional information about the traffic that has been observed over time, i.e., its trends. For example, historical data on the duration of individual CETP sessions and information on sent and received packets within the session may allow for their analysis and, as a result, for the detection of anomalies. The scenario with data exfiltration detection will be discussed later in Sections 6 and 7. In the RGW scenario, CETP is not used and the initial communication is taking place using only DNS protocol. This case can be supported by the default Passive DNS configuration. Exemplary samples from the Passive DNS/CETP solution are presented in Tables 5 and 6. It may be concluded that storing data in this form is really helpful as every step of the negotiation can be checked in details and any discrepancies can be quickly identified. In Table 5 we can observe typical information related to the CETP negotiation process, policies exchanged, and how long such process took. This data also has a reference to the DNS data collected in the previous step of communication establishment process so there is an access to the full scope of valuable data. In Table 6 another example is visible where mapping of Passive DNS record to information about the session is enclosed. In general, the process of analysis can be very laborious if the investigator is provided only with raw PCAP data. In this case, the investigator has to look for the initially exchanged DNS traffic and based on that build relationships with the CETP traffic which must be performed manually. Carving out the statistical information is also not an easy task. Thanks to the extension such as Passive DNS module, the investigator is provided with the full perspective of the whole communication process in a simple way and has enhanced visibility of everything what is happening on the wire. The relationships between different phases are already built and this can greatly support analyses of the detected anomalies or other malicious activities.

**Table 5.** Exemplary output of the Passive DNS with CETP support–Policies.

| Negotiation Start | Negotiation End | Offer Sent |
|:---:|:---:|:---:|
| 1,588,322,844 | 1,588,322,846 | Id.fqdn, ctrl.cesid, rloc.ipv4 |
| **Answer** | | |
| 172.217.20.174 | | |
| **Require Sent** | **Available Sent** | **Offer Received** |
| Id.fqdn, ctrl.cesid, rloc.ipv4 | Id.fqdn, ctrl.cesid, rloc.ipv4 | NULL |
| **Require Received** | **Available Received** | **Passive DNS Record Number** |
| Id.fqdn, ctrl.cesid, rloc.ipv4 | Id.fqdn, ctrl.cesid, rloc.ipv4 | 2 |

**Table 6.** Exemplary output of the Passive DNS with CETP support–Session Parameters.

| SST | DST | Rate Limit | TTL | Passive DNS Record Number |
|:---:|:---:|:---:|:---:|:---:|
| 234 | 432 | 1,000,000 | 86,400 | 2 |

## 5. Risk-Based Analysis Engine for the Passive DNS/CETP

*Risk-Based Analysis Engine*

Passive DNS/CETP module analysis engine operates in real-time on the data collected by the module. It also enriches them with data obtained from the threat intelligence. The engine processes the data collected by the Passive DNS/CETP module and, on their basis, prepares new data in the form of a report that is more beneficial for the analyst or a different automation engine. The concept is shown in Figure 11. The computational complexity of the proposed solution is O(n), where n is just the number of bits in the packet. This is because in order to extract all the required information from the DNS/CETP packet, the Passive DNS/CETP module has to look through all the bits of the packet one by one. The solution is not difficult to use as it has very clear interface. Each of the elements of the proposed solution in Figure 11 is described in details below:

1. CES-A: CES firewall implementation. Responsible for blocking unwanted traffic and forwarding benign traffic by inspecting individual data flows. Negotiates relevant policies with other CES firewalls necessary to exchange information.
2. Passive DNS/CETP module: a module that looks for DNS and CETP packets in traffic going through the CES firewall. Its most important functionality or the main algorithm of operation is the extraction of information from both types of packets. These are also the input data of the module. After extracting the information, it places it in the database, which makes access to it easier. So this is the output of this module.
3. Database: the database is the main place for storing information on DNS and CETP traffic. Information obtained from the Passive DNS/CETP module is provided at the database entry. The database stores this information and enables quick access to it. The data is stored on the per-packet principle, i.e., the main structure of each entry in the database is simply a packet.
4. Risk-based Analysis Engine: an engine that polls the database using many different queries while extracting information such as last seen and active data flows or information about individual DNS and CETP sessions. The engine communicates with the SIEM platform in order to deliver and receive the most important information about each session. In addition, he also contacts Threat Intelligence in order to enrich the decision-making process with additional information on IP addresses or domain names.
5. SIEM platform: a set of services and tools offering insight into the overall information security of an organization. Offers Real-time visibility across information security in the organization, event log management along with the consolidation of logs from multiple sources and their correlation, or the creation of automatic event-driven

notifications. Here, it is integrated with Risk-based Analysis Engine to exchange the data.

6. Threat Intelligence: evidence-based knowledge. This is primarily about context, mechanisms, indicators or action-oriented advice. This knowledge concerns an existing or emerging threat. Threat intelligence can be used to make decisions about a specific data flow. Here, this knowledge supports the risk-based analysis engine decision-making process.

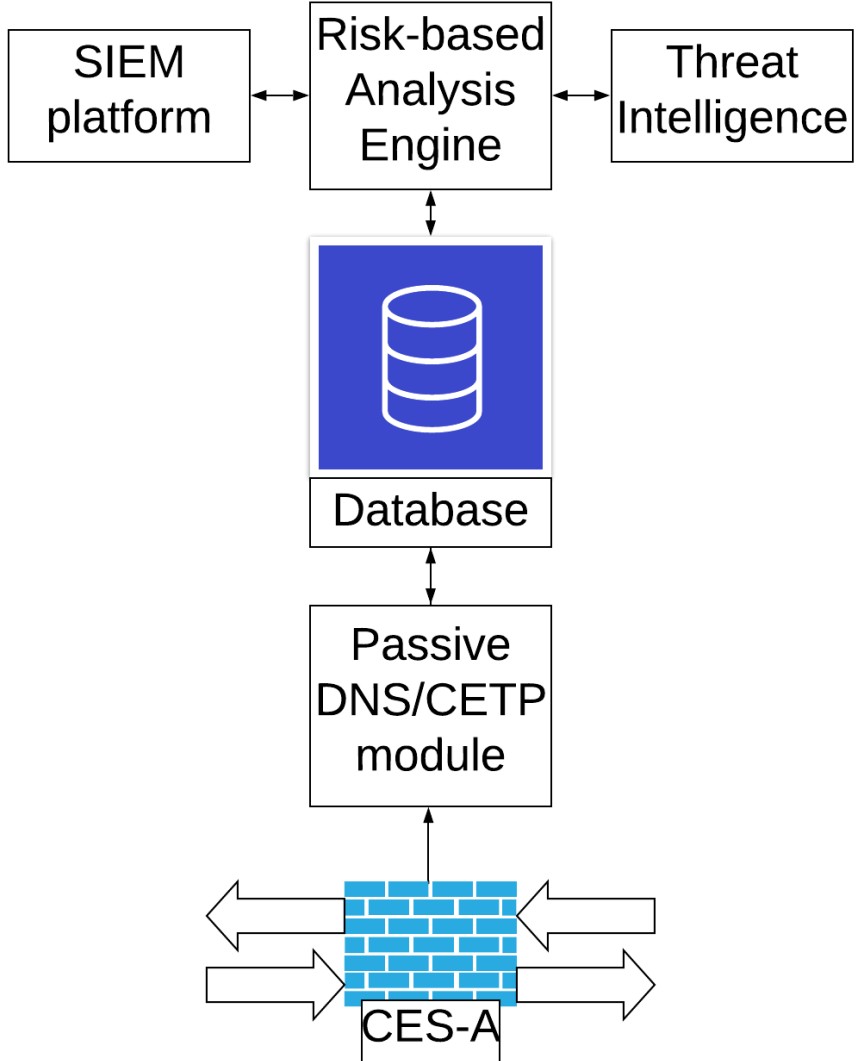

**Figure 11.** Model of Risk-based Analysis Engine with Passive DNS/CETP.

Passive DNS/CETP module observes all traffic that passes through the firewall in both directions, extracts information about DNS and CETP traffic as well as metadata describing traffic such as connection duration, amount of data sent or received. Obviously the type of information extracted can be selected based on the business requirements. This module then places the collected information in the database. The risk based analysis engine operates on the data stored in the database and, on that basis, analyzes whether any anomalies have been observed. Anomaly detection is based on an automatically determined decision threshold. Based on this threshold, the analysis engine assigns an appropriate risk score and prepares the priority list.

The main implementation goal of the data analysis engine is to detect anomalies in the network traffic. The effectiveness of the available anomaly detection solutions in data traffic

vary and the number of proposed solutions is significant and diverse as it was mentioned in subsection Related Work. The analysis engine for the Passive DNS/CETP module is therefore based on a completely new assumptions which, to the best of our knowledge, have not been widely used so far. The main novelty of the proposed approach is that it operates on the risk and returns the priority list of flows that need attention. It is the risk that is a key factor. The risk-based solution looks for anomalies in the behavior of various elements on the transmission path and according to the detected anomalies, it assigns an appropriate risk score to each of them and sums them all together to form overall risk rating for the communication session and put the session in the appropriate place on the priority list which is accessible by security analyst. In this paper, we apply this concept for the CETP sessions, but the data analysis engine can modify the risk score of the host, network, service, etc.

The detection process is based on the selection of appropriate scenarios consisting of 'domains' (i.e., areas of observation) that the module should track and in which to look for anomalies. For example, one such scenario is the detection of data exfiltration. We can look for data exfiltration using many domains. The tests that were used to create this scenario for the purposes of this paper will be presented later. Obviously, the risk score returned by individual test can be assigned appropriate weights, which seems obvious if, for example, a communication session towards the domain with the worst possible reputation is discovered in the network.

It is also obvious that the higher the risk score assigned by the particular test the more the determined value exceeds the decision threshold or deviates from the mean or the median. However, in this paper, we assume a binary approach, where the risk score is equal to 1 and this corresponds to detection, while in the absence of detection, the assigned risk score is equal to 0.

The information about the reputation of the certain domain or information from the threat intelligence in general is not always perfect so such traffic is usually blocked or just passively monitored and alerted. In any case, the assigned risk score should be relatively high. Such a risk-based approach provides an enormous benefit of automatically prioritizing observed traffic that has violated the relevant decision thresholds.

If the certain features of the communication session exceed many of decision-making thresholds, then it will receive a very high risk score and will appear in the first positions of the prepared report. This allows the analyst to focus the available resources on the most safety-critical communication sessions. The most obvious case seems to be data traffic that will breach the thresholds set in all tests used in the scenario. It is also obvious that there will always be a malicious session that can stay below any set of thresholds. However, the more such tests we use in a particular scenario, the more difficult it is for an attacker to bypass them unnoticed. Even if the malicious actor would send one bit in each newly created session for the purpose of data exfiltration to a domain that is considered safe, the tests examining the amount of information sent would not indicate any anomaly related to the data exfiltration. However, the test that would examine the number of sessions created would indicate an anomaly and thus increase the risk score. The situation would therefore be prioritized and shown in the report.

The obvious situation is also the one in which the network traffic may be or may be not completely benign. An example of this is the case of synchronizing data from a computer with the cloud or sending an email with a large attachment through the service offered by a large service provider. Such approaches can be also used by malicious actor to exfiltrate data. The risk-based solution will assign an appropriate risk score for such traffic, the higher the more decision thresholds that the traffic exceeds in individual tests.

The completely benign situation or not will, therefore, appear in the report and may even be placed on the top of the list, because the engine does not possess all the information. It is for this reason that the data analysis engine for the Passive DNS/CETP data module enables its integration with central Security Information and Event Management (SIEM) platform. The analyst reviewing information about individual communication sessions,

starting naturally with those with the highest risk score, can correlate information about them with logs from other sources, e.g., from the machine participating in the communication. If logs from another source indicate that a suspicious process running on the device is responsible for the traffic or the user's activity at that time was suspicious, it may be concluded that data exfiltration actions indeed took place.

The data analysis engine has an interface that allows you to enrich the Passive DNS/CETP module with additional information about the risk. If a machine has been identified that generates a lot of suspicious traffic, it seems reasonable to inform the module's data analysis engine about this fact. In result, the engine will raise the risk score for the host and thus in each generated report, any sessions of that host will appear on the higher positions. The coupling between the systems brings a measurable benefit. The data analysis engine allows, of course, to manually place individual elements involved in the communication on a black, grey or white lists. The black list blocks the element completely, the white list absolutely allows the traffic, and the grey list allows partial activity taking into account defined scenarios, tests and security policies. It depends on the configuration whether elements placed on the lists will be included in the report.

## 6. Experimental Implementation and Methodology

The test-bed used in the experiments is shown in Figure 12. It supports CES-to-CES communication. As it is seen, both CES devices have additional modules responsible for Passive DNS/CETP operations. Each module implemented on the CES device identifies DNS and CETP traffic in the passing data, extracts the interesting packets, and inserts the desired information in the database. The module is also responsible for building appropriate relationships between DNS and CETP packets so it can be easily used for the investigation later.

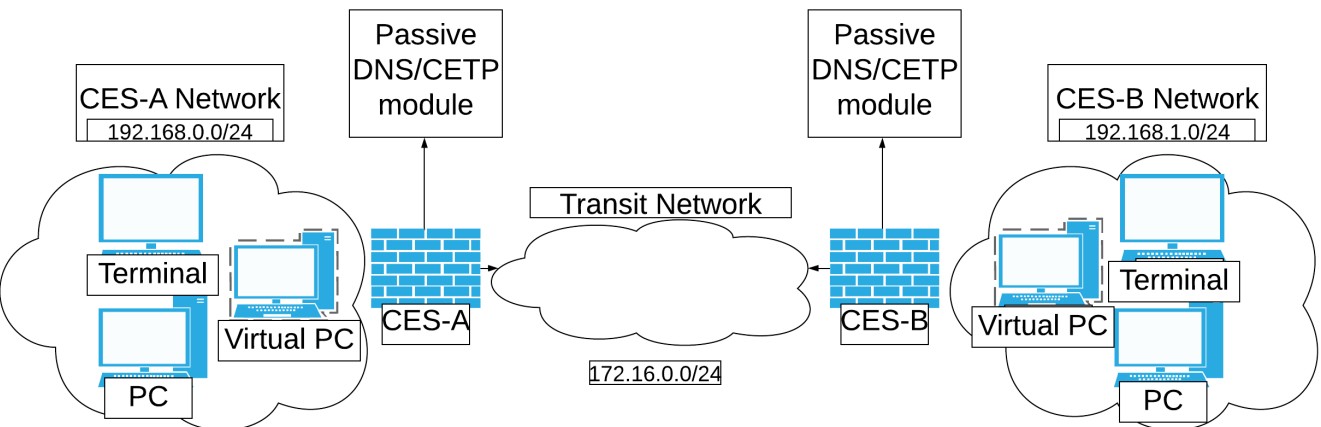

**Figure 12.** Implementation of the CES-to-CES test-bed.

The test-bed was built using two laptops (Microsoft Surface Pro X SQ1 8 GB 256 GB SSD W10) where each was placed in CES-A and CES-B networks respectively. The laptops were used for generating the traffic and the whole communication process. It was decided to generate traffic representing typical computer use, i.e., the user uses a web browser, antivirus programs and installed applications that generate traffic to the cloud to synchronize the data. The generated data traffic contained a total of 141 MB of data. This comprised 1264 TCP sessions and 15 UDP sessions, including 1 malicious UDP session and 2 malicious TCP sessions. The entire test lasted 17 min. The decision to choose this type of traffic is justified by the popularity such user behavior. The generated traffic is characterized by a variety of different flows. Traffic has a wide spectrum of flows, starting from those with short and low-speed connections, such as a browser-generated connection with a request for a specific page, or those with long and generating a lot of traffic flows, such

as data synchronization with the cloud. The experiments lasted 30 min. The CES-based cooperative network firewalls were developed using Raspberry Pi 4B platforms, each with 8 GB of RAM. Both CES devices were directly connected to each other. The whole software that forms the Passive DNS/CETP was implemented in the C programming language. The choice of this language was justified by the reasonable speed of the operation which was one of the main assumptions of the prepared module. The database engine chosen to store all information was MariaDB ([mariadb.org](mariadb.org), acceseed on 15 March 2021).

The first experiment consisted in preparing sample data sets in files of various sizes, which were used by the end stations to generate desired traffic. After successfully transmitting/receiving all packets from a particular file, the performance of searching for a specific CETP session was checked by comparing the speed of information extraction from the database with the speed of popular packet search tools such as TShark and Ngrep. The experiment was repeated 1000 times, inspecting a different CETP session each time.

As the Passive DNS/CETP module collects historical data from various fields and metadata, it has been decided to conduct the second experiment using the risk-based data analysis engine to obtain additional information and, as a result, demonstrate anomaly detection in data traffic. The experiment demonstrates how the potential data exfiltration in the networking environment can be detected. In order to conduct the research, the files with packets generating various CETP sessions on CES firewalls were prepared. These files were used by the end-stations to simulate different communication processes so the Passive DNS/CETP module was able to collect historical information about sent and received data in particular CETP sessions. Based on these historical data, an attempt was made to detect all potential data exfiltration attempts using the adaptive threshold. The algorithm used in the implementation of the adaptive threshold was Cell-Averaging Constant False Alarm Rate (CA-CFAR) [58] which is probably the most widely used CFAR detector. It was used as it can adapt to a constantly changing networking environment.

## 7. Results

In this section we present results for the two designed modules introduced in this paper. First, we showcase the results related to the evaluation of the performance of the Passive DNS/CETP module. Then the efficiency of the risk-based detection module is determined.

### 7.1. Performance of the Passive DNS/CETP Module

For this part of the research we wanted to evaluate the performance of the Passive DNS/CETP module when compared to the other commonly used search tools such as TShark or Ngrep. We ran tests for three different file sizes: 10 MB, 100 MB and 1 GB. Each individual file was searched 100 times, each time for a different CETP session. The statistical data collected were the averages of the search time and the standard deviations.

For the 10 MB file, TShark searched for particular CETP sessions in average of 1.193 s (standard deviation of 0.4 s), Ngrep searched in average of 64 ms (standard deviation of 26.13 ms) for Passive DNS/CETP it took only 3 ms (standard deviation of 0.47 ms). This file contained 92 k packets. For 100 MB file and 925 k packets, it took 10.958 s (standard deviation of 4.08 s) for TShark, 1.521 s for Ngrep (standard deviation of 0.53 s) and 29.7 ms for Passive DNS/CETP (standard deviation of 0.14 ms). The last file tested, 1 GB file with 9262 k packets, took TShark 109.458 s (standard deviation of 40.82 s), for Ngrep it took 14.740 s (standard deviation of 5.71 s) and for the Passive DNS/CETP only 315 ms (standard deviation of 3.01 ms). The results for particular PCAP files are shown in Figures 13–15, respectively. It is clearly visible that the Passive DNS/CETP is much more faster than two other solutions. The Passive DNS/CETP offers much better results because the data has been already stored in the database. The comparison of the obtained results is presented in Table 7.

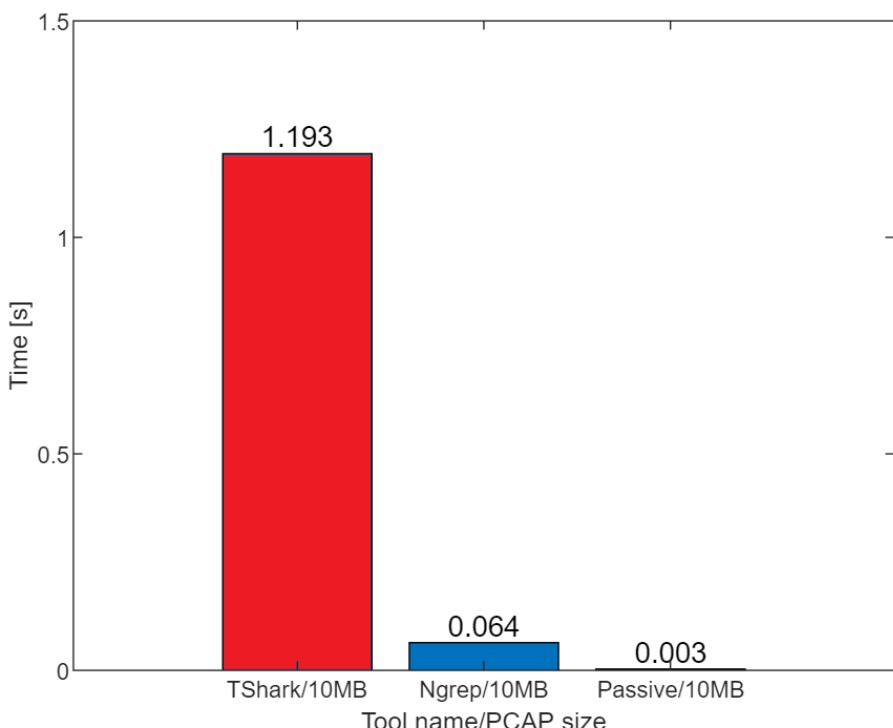

**Figure 13.** Comparison of the execution time for different tools and 10 MB PCAP file.

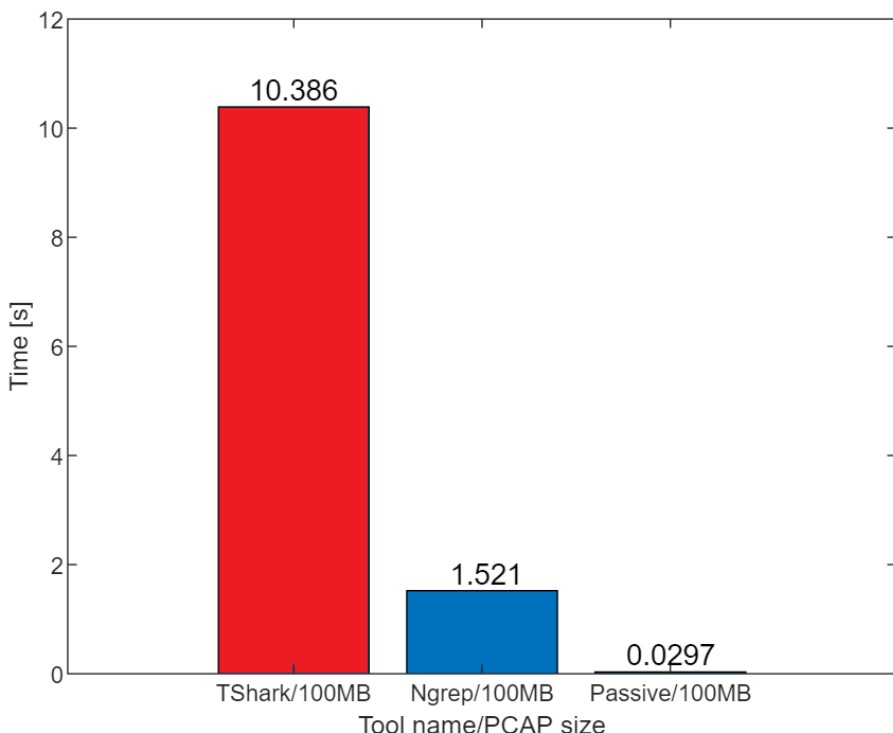

**Figure 14.** Comparison of the execution time for different tools and 100 MB PCAP file.

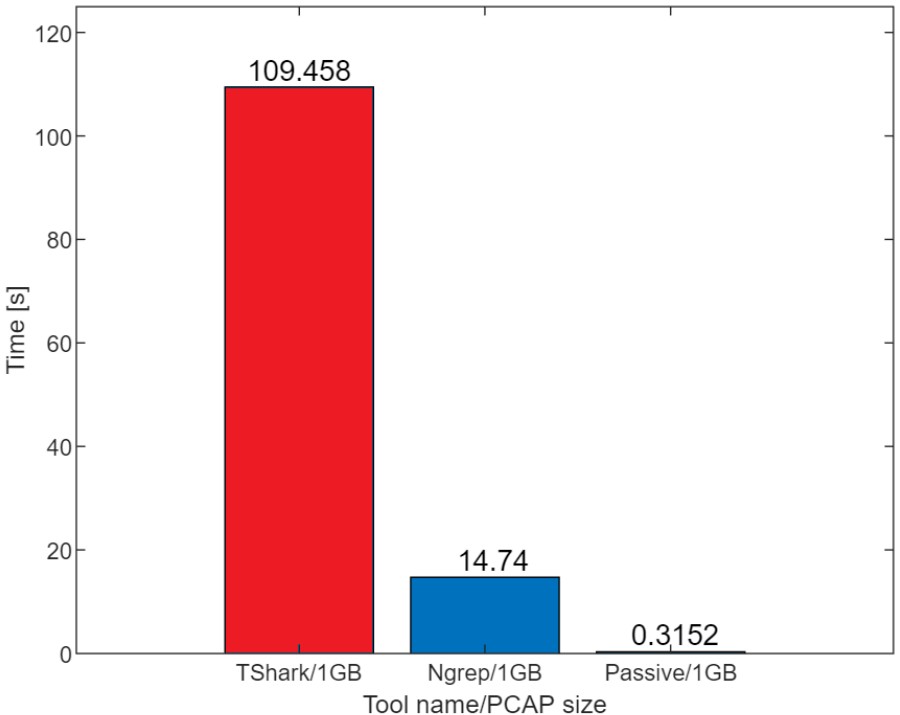

**Figure 15.** Comparison of the execution time for different tools and 1 GB PCAP file.

**Table 7.** The state-of-the-art comparison between the approach and the execution time for different PCAP sizes.

| Approaches/PCAP Size | TShark | NGrep | Passive |
|:---:|:---:|:---:|:---:|
| 10 MB | 1.193 s | 0.064 s | 0.0030 s |
| 100 MB | 10.958 s | 0.587 s | 0.0297 s |
| 1 GB | 109.458 s | 14.740 s | 0.3152 s |

*7.2. Efficiency of the Risk-Based Detection Module*

The experiments related to determining the efficiency of the risk-based data analysis engine for the Passive DNS/CETP module were evaluated for the data exfiltration scenario. For this purpose, data packets simulating ordinary user's activity in the network using a browser and popular applications and services were generated from the end host.

The generated data traffic contained benign traffic (marked in blue) and also contained three completely different and relatively hard to detect malicious sessions that were responsible for data exfiltration (marked in red):

- a session which very slowly exfiltrated a sensitive file to the newly created domain imitating typical TLS session,
- a session which exfiltrated a sensitive file by uploading it to relatively trusted website, and,
- a session which exfiltrated a very small sensitive file containing passwords by dividing them to the smaller parts and sending in different sessions to the domain with a slightly weaker reputation.

As a result of the generated traffic, the appropriate CETP sessions were created on the CES firewalls and the information about the sessions was collected by the Passive DNS/CETP module. During the generation of packets in the network, the Passive DNS/CETP module and the data analysis engine were queried for a report on the last 100 active sessions monitored by the module. For this time window, CA-CFAR algorithm set the detection threshold using 22 training cells and 2 guard cells. The values of training

cells and guard cells were selected empirically by checking how the decision threshold influences detection.

The study of the impact of the other parameters of the CA-CFAR algorithm on the detection performance for configurations: 10 + 2 (10 training cells and 2 guard cells), 40 + 2, 10 + 4, and 40 + 4. The designated decision thresholds are shown in Figures 16–19, respectively.

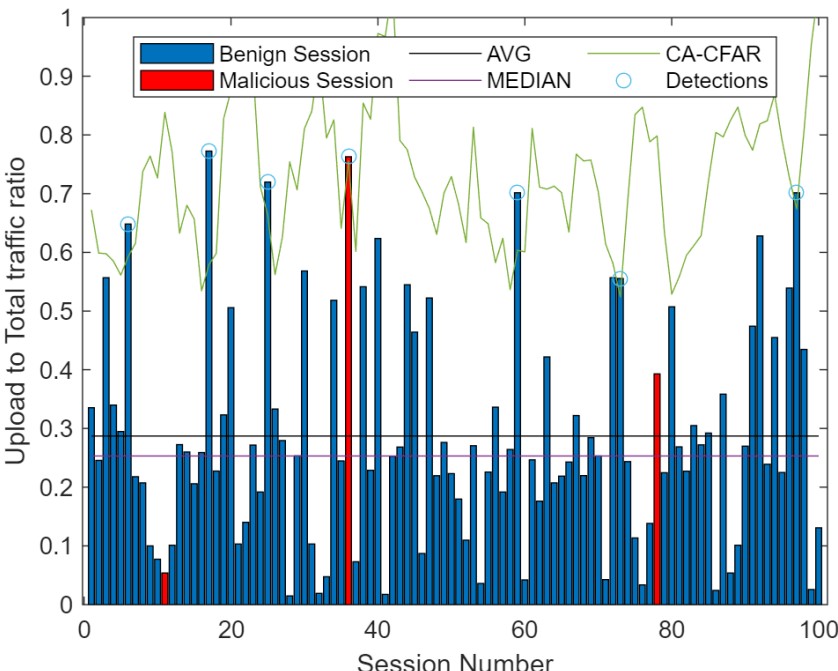

**Figure 16.** The designated threshold for 10 training cells and 2 guards cells. Upload ratio.

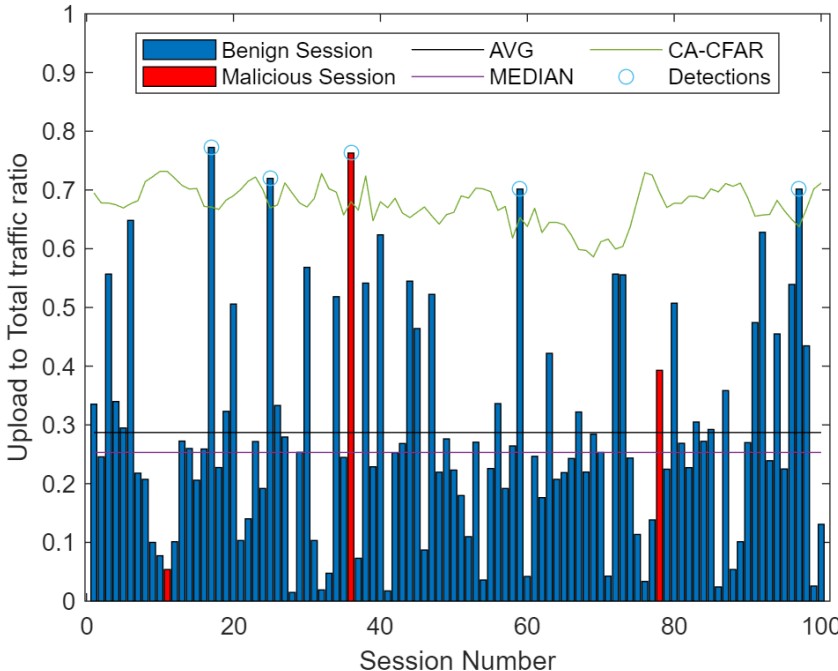

**Figure 17.** The designated threshold for 40 training cells and 2 guards cells. Upload ratio.

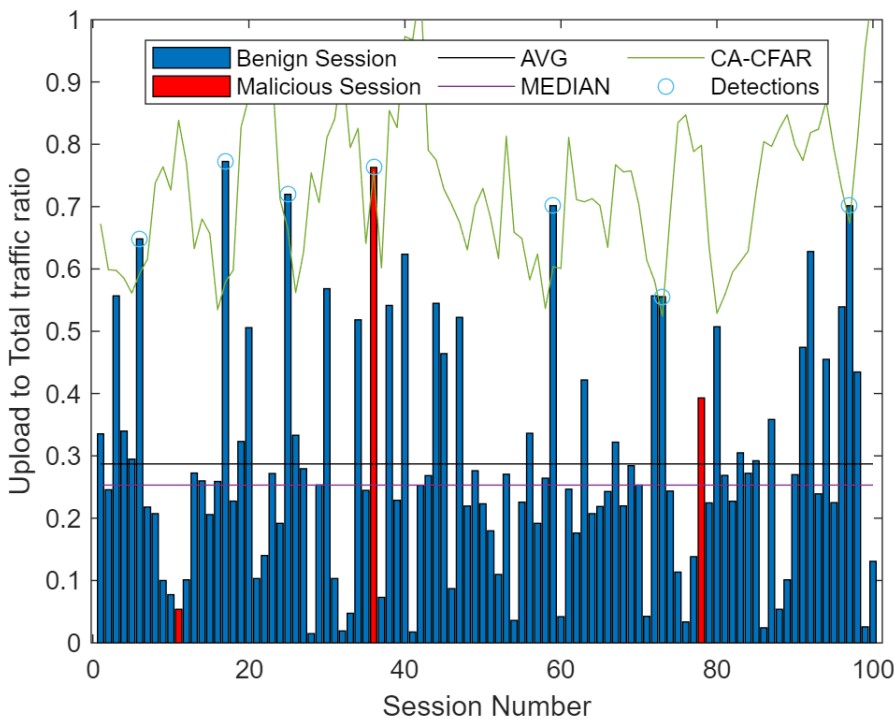

**Figure 18.** The designated threshold for 10 training cells and 4 guards cells. Upload ratio.

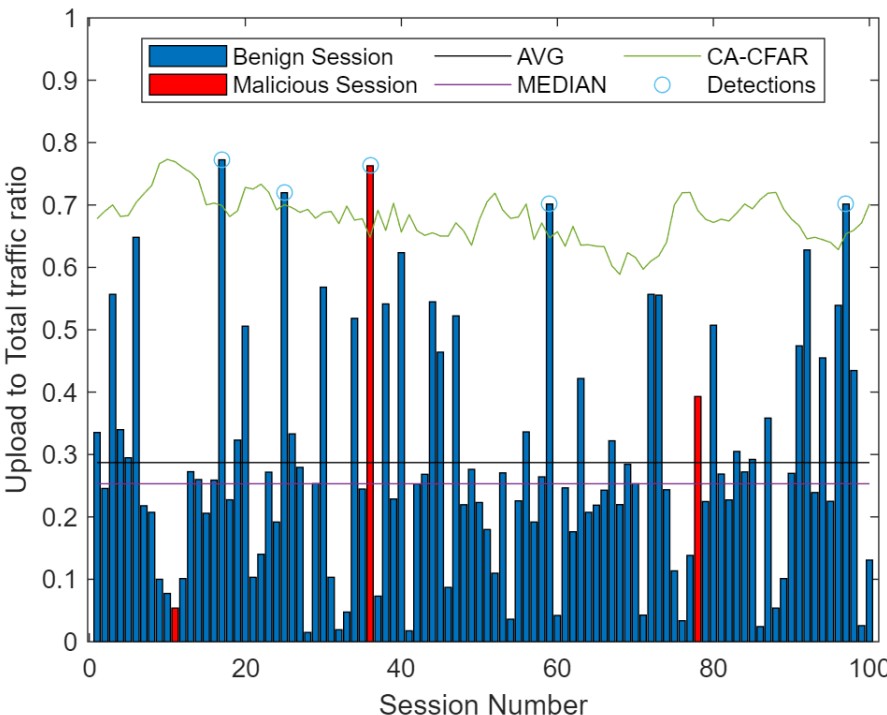

**Figure 19.** The designated threshold for 40 training cells and 4 guards cells. Upload ratio.

As can be seen from the figures, lower values of training cells result in the decision threshold being more aggressive. On the other hand, higher values of training cells flatten the chosen decision threshold, so the detection concerns only the highest values from the chart. As for the values of guard cells, their impact on detection is relatively small. Guard cells only gently influences the shape of the decision threshold.

The report contained information on the detected anomalies in individual sessions and the risk score assigned to them. Note that the most important information contained in the report is presented below and properly discussed.

The study began with checking the anomaly detection on the basis of the obtained reputation scores assigned to the observed domains. This was an obvious choice due to the widespread use of this approach. As we can see in Figure 20, two attempts to exfiltrate the data were detected.

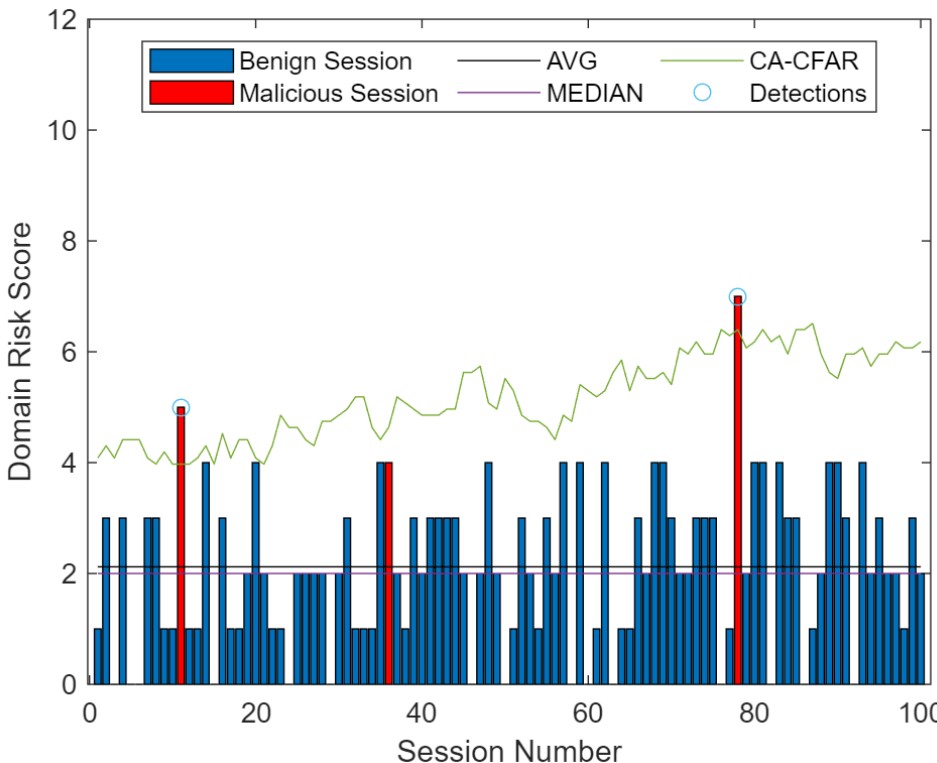

**Figure 20.** Detection based on the information about domain reputation.

During the first session data was sent to a recently registered domain, so the analysis engine had no information about it neither threat intelligence so the engine assigned it a neutral reputation of 5 (where 0 is a trusted domain, 10 is a fully malicious domain, assigned appropriately by risk analysis engine). These types of new domains are often used by malicious actors (but obviously this is not always the case). Due to the good reputation of neighboring sessions, the session to the new domain exceeded the decision threshold. Within the second session data was sent to a domain commonly considered as malicious. The third exfiltration attempt (the one in the middle in Figure 20) was not detected due to its connection to a relatively trusted domain. Due to the effectiveness of finding anomalies based on the reputation of the domain, the assigned risk scores have been scaled by 5. The first session received a risk score equal to 1 times 5 so 5, and the second session risk score equal to 1 times 5 so again 5. None of the other benign sessions was classified as an anomaly, so the risk score for them remained at 0 at this point.

The next step was to check how much data was sent in each session (Figure 21). This approach is also quite straightforward as we are trying to detect the data exfiltration scenario. The results here, however, are no longer as desirable as in the previous case. We can see that two exfiltration sessions have been detected again, but we also have quite a few detection alarms for the benign sessions–10 to be exact.

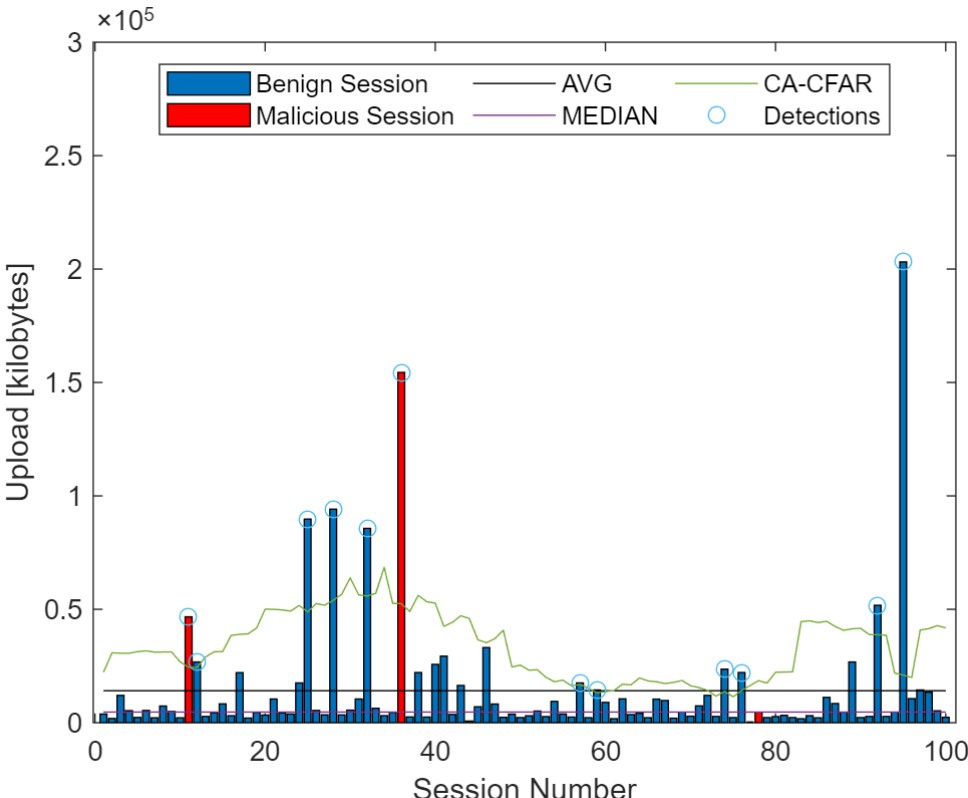

**Figure 21.** Detection based on the information about upload size.

They were detected because they were significantly different in the size when compared to the volume of data sent from other sessions. After analyzing the data from the report, it turned out that these benign sessions were responsible for synchronizing data with the cloud, sending data to social networks and mostly traffic generated to analytical websites collecting information about users when running the browser. We can exclude some of these sessions from this test by entering them into the appropriate lists, but for our case, let us assume that all detected events receive an additional point to the risk score. Each added point causes the flow to appear higher on the priority list so that the security analyst can check the flow faster by browsing the priority list from top to bottom and react faster. After all, malicious actors could use these flows to exfiltrate data, too. The added point to the risk score is therefore entirely justified.

The next experiment was aim to verify the ratio of the data sent to the total amount of data exchanged (Figure 22). For the typical exfiltration attempt, this ratio should be large, because in the end we send larger packets than the packets we receive as confirmation. In this case, there was one correct detection of malicious activity and four false detection of the benign sessions. These four benign sessions are, respectively, an advertising service, a social networking site, an information search portal and its analytic subservice. Of course, if we assume that the analyzed flows included traffic such as video streaming or SFTP transmission, the traffic, if it were in the vicinity of low-value flows, would also be detected in this domain. However, there will be no detection if high-value flows in this domain are closely adjacent to each other. As such services can be abused for data exfiltration, so their detection and insertion in the appropriate place in the priority list may be justified. Risk and the priority list are the keys.

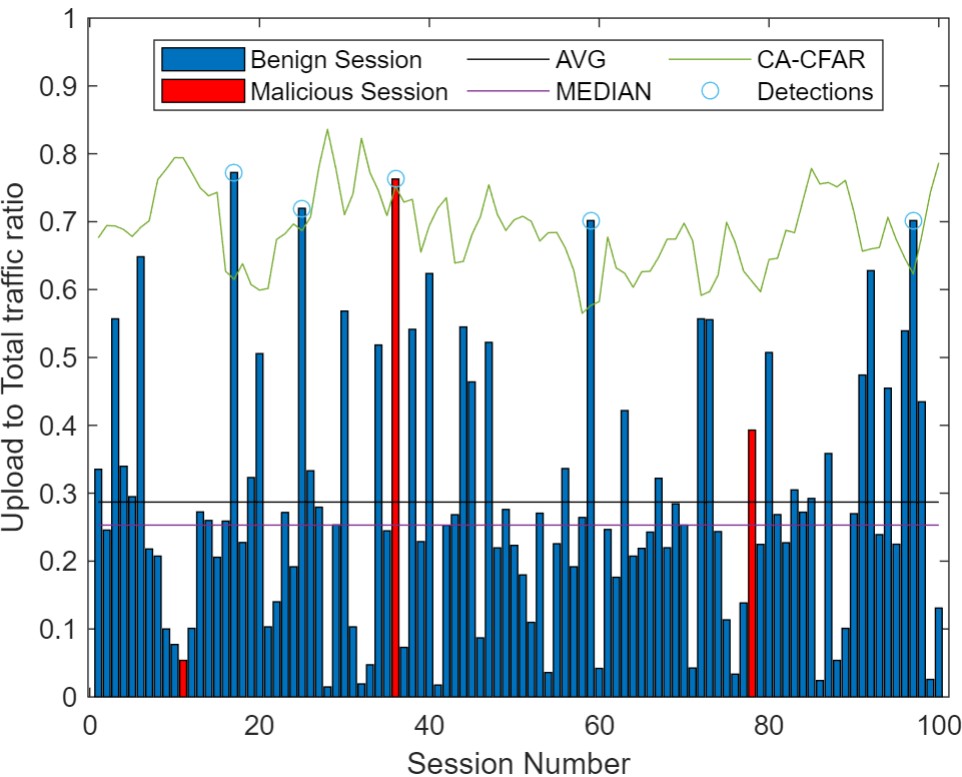

**Figure 22.** Detection based on the information about upload ratio.

The next test in the exfiltration detection scenario used analysis of the session duration parameter (Figure 23). It has been assumed that long sessions usually transfer a lot of data. This experiment successfully pointed to only one malicious session but also identified incorrectly 11 benign sessions. After analyzing all benign sessions, it turned out that some of them are websites responsible for analytic and collecting information about the users, but the rest are ordinary popular websites for which sessions were maintained due to user activity. The report generated at the beginning of the study showed a significant share of the TCP Keepalive packets for sustaining session duration. Thus the obtained results do not look promising. Again, the data can be exfiltrated in a variety of ways. The attacker may place them on any website. That is why, each detection alarm gains an additional point to the risk score.

The goal of the final experiment was to check how many different flows were observed in each CETP session (Figure 24). To avoid detection, the malicious actor may break up the file into smaller parts and send them in separate sessions that mimic a normal traffic. As a result, one malicious session was detected. The majority of the rest of the traffic was based on one flow to the server. Several sessions consisted of two or three flows, and these were flows mainly to the same analytic service providers. These sessions came from various websites. As before, one detection means one more point on a risk score.

The console window showing 10 flows with the highest risk is shown in Figure 25. It contains the basic information about the flows, i.e., IP addresses, ports involved in the information exchange or the value of the assigned risk based on the performed experiments. The console panel summarizes the flow studies in the various tested domains and provides the security analyst with an overall view of what is currently happening on the network. The information in the window is refreshed in the same way as information is displayed using the top command.

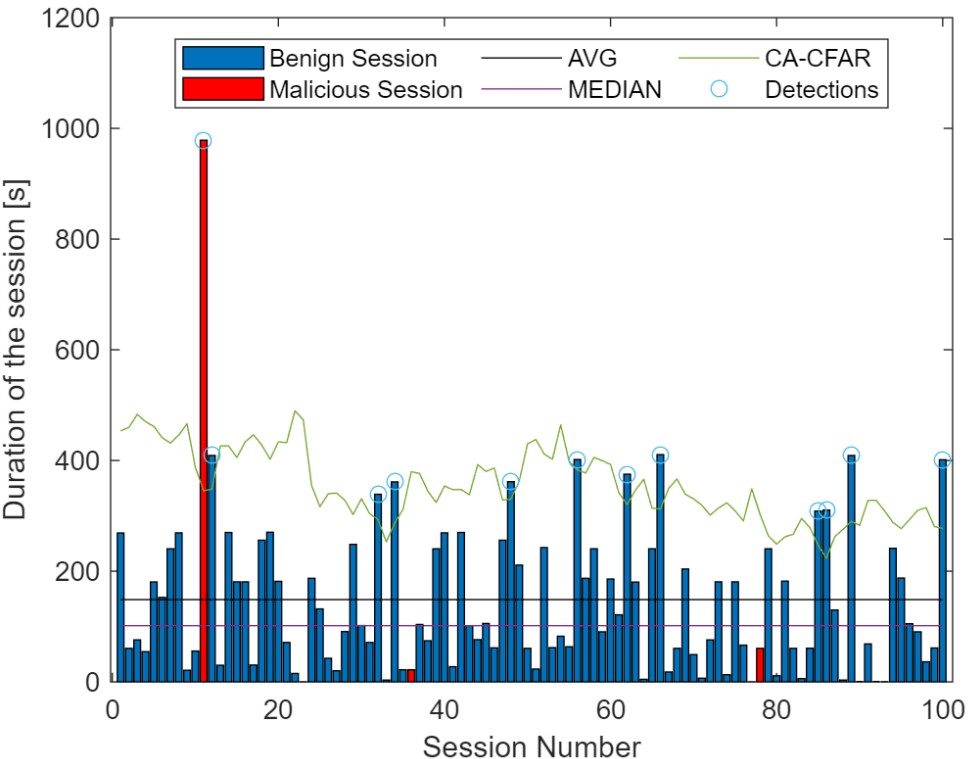

**Figure 23.** Detection based on the information about session duration.

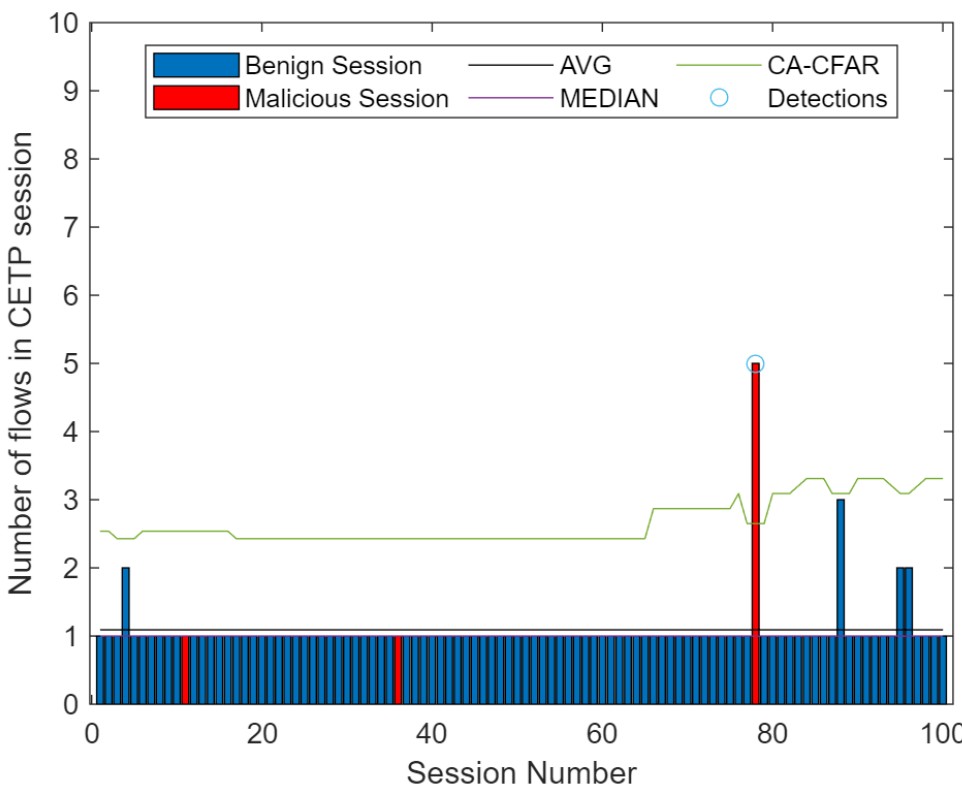

**Figure 24.** Detection based on the information about number of different flows seen in the particular CETP session.

**Figure 25.** Console window - Top 10 Highest Risk Score Flows.

Summing up the results of the entire study, it is visible that the biggest advantage of the risk-based approach is that it prioritizes potentially malicious data sessions according to the highest risk score assigned. In the result, it allows resources to be focused on them, thereby detecting and responding faster, which is highly desirable in a complex environment.

Moreover, it enables the detection of the most advanced types of data exfiltration. This study deliberately selected exfiltration cases that were relatively difficult to detect and operated on small file sizes. Obvious cases of data exfiltration, where the session lasts a long time and sends a lot of information, were omitted from the study, because such sessions will have a very high risk score by definition.

The conducted experiment definitely detected a malicious session that slowly exfiltrated the file with sensitive data while imitating a TLS session. This session received a total risk score of 7. The experiment also detected a session in which the malicious actor, to avoid detection, split the sensitive files into many smaller parts and sent them to a server with a slightly weaker reputation in various flows. This session received a risk score equal to 6. The last session that sent a very small file with passwords to the social networking site received a risk score equal to 2. Analyzing the rest of the benign sessions, two of them responsible for communication with well-known websites received a risk score of 2 and twenty received the risk score equals to 1. It is also worth reminding that in the experiment, to simplify the operating mechanism, we adopted a binary approach to calculate the risk score. The graphs show that the malicious sessions significantly deviate from the threshold values or mean/median values in many domains.

The essence of the study is that detection in different domains can have different benefits. There are domains where the investigation of individual attacks has the advantage of a low false positives and a relatively large number of true positives, but there are also domains where detection is difficult and the overall number of detections is high. The study of traffic characteristics in many domains may lead us to obtain a priority list, where individual flows are organized according to the assigned risk from different domains and thus a better use of resources in the investigation.

## 8. Case Study

Suppose we reveal that a new malicious domain used by the attacker in the DNS technique, called Fast Flux, appears on the Internet. We would like to see if the hosts on our network have contact with it. Thanks to the fact that the Passive DNS/CETP module collects the most relevant information about the observed sessions and stores historical data, we have complete visibility whether any such activity occurred in our networking environment.

Using a single query to the module, we obtain information on all hosts that contacted the malicious domain. In addition, we obtain information on when the relevant communication sessions occurred. In this way, we can very quickly identify our 'patient zero' and the time of its first contact with the malicious domain.

By integrating this information into the SIEM platform, we can effectively correlate the activity of this host based on logs from other sources. It turns out that the host has previously contacted another relatively trusted domain and downloaded an infected file from it. All hosts can be quarantined and moved to a different subnet until the damage caused by the malware is repaired. Passive DNS/CETP module and its risk-based data

analysis engine can be informed that the hosts have been infected and their risk score should be raised to a sufficiently high level so that all suspicious activity of the infected hosts on the network is listed relatively high in the report and information about additional malware activity could be collected. After gathering as much information as possible about the malware, we can respond to it much more effectively.

## 9. Conclusions and Future Work

In this paper, we showed how the implementation of the dedicated Passive DNS/CETP module can improve the performance and the overall investigation process at the cost of initial database creation. We showed that the Passive DNS/CETP functionality offers a much better approach when searching for a particular session.

We also introduced the concept of risk-based analysis engine and its detection capability. Our main contribution, based on the experimental results, was that we evaluated the performance and showed what were the execution times for the Passive DNS/CETP and other commonly used search tools. It turned out that the execution times for the new approach were much better than the ones for TShark or Ngrep. Passive DNS/CETP searched for the particular session 350–400 times faster than TShark and 20–50 times faster than Ngrep.

Moreover, by defining the data exfiltration scenario and its tests in the risk-based data analysis engine, we examined its effectiveness in detecting anomalies and prioritizing potentially malicious sessions, which would allow to focus resources on the most critical flows. It had been evaluated that the risk-based approach enables the detection of relatively difficult to detect exfiltration methods. Considering the received experimental results, the Passive DNS/CETP and the risk-based analysis engine were novel approaches worth implementing as they support a better investigation process, allowing to search for particular sessions orders of magnitude faster than the other approaches and risk prioritization of the data traffic, which translates into a more efficient response to the most security-critical communication sessions. Both proposed solutions enhanced the overall security of the environment.

Passive DNS/CETP module and its risk-based data analysis engine were, to the best of the authors' knowledge, relatively new approaches to anomaly detection in the networking environment. These were certainly topics that should be better researched. The main goal of our future research aims to define new scenarios and new tests and to examine how the risk-based detection performed for these cases. In particular, we would investigate whether an increase in the number of scenarios and tests translates into better performance. It would also be good to examine the effectiveness of the solution using other algorithms for determining the decision threshold and their various configurations.

**Author Contributions:** S.N. carried out the experiment. S.N. wrote the manuscript with support from W.M.; S.N. fabricated the Passive DNS/CETP and Risk-based Analysis Engine samples. W.M. helped supervise the project. S.N. conceived the original idea. All authors have read and agreed to the published version of the manuscript.

**Funding:** This research received external funding.

**Data Availability Statement:** Customer Edge Switching files can be found on https://github.com/Aalto5G/CustomerEdgeSwitching. The data that support the findings of this study are available from the corresponding author, Slawomir Nowaczewski, upon reasonable request.

**Acknowledgments:** This work was undertaken under the "Context-Aware Adaptation Framework for eMBB services in 5G networks" project supported by the National Science Centre, Poland, under the grant Nr. 2018/30/E/ST7/00413.

**Conflicts of Interest:** The authors declare no conflict of interest.

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
