# Peer review of "Improving Security of Future Networks Using Enhanced Customer Edge Switching and Risk-Based Analysis"

_electronics, doi:10.3390/electronics10091107_

Round 1

Reviewer 1 Report

In this paper, the authors show how the Passive DNS can be used to further improve security of this solution.It is poor. I have the following questions:

(1)The title of the article seems like a long sentence.

(2)They should clearly state the innovations from the application point of view and they should clearly define which are the innovative features of their proposal with respect to adopted logics.

(3)The overall organization of the paper is very poor. Why proposal part is too short and lack of explanation about technical terms.

(4) The authors should compare the performance of the proposed method with other new schemes.

(5)Is the complexity of the proposed scheme very high? Is it difficult to use in practice?

(6)The authors also should compare the computational time complexity of the proposed method with other new schemes. They should clearly state the point.

(7)The use of the present tense in Conclusions is not correct.

Reviewer 2 Report

Currently, information security is the most important requirement for an infocommunication system. The trends in the development of communication networks are aimed at increasing their capabilities in terms of information delivery, increasing bandwidth and penetration into various spheres of human life. General development trends are such that they increase the availability of information, which is undoubtedly a very favorable factor stimulating the development of industry and society. The flip side of this process is the growing danger of using the possibilities of access to information to commit destructive actions. In this regard, ensuring information security is the most important direction in the development of a modern infocommunication system.
The aim of this work is to improve information security, in this sense, this work is relevant, and its results can be used in modern and promising communication networks.
The authors of the paper propose using the DNS / CETP mechanism at the user boundary to identify suspicious traffic. According to the authors, the mechanism proposed by them has advantages over the known mechanisms, which can be numerically estimated by the time required to identify threats.

The authors performed an in-depth analysis of publications on the topic of the work and presented its results (Table 1), which clearly demonstrate the directions of research in this area.
The authors have done a lot of experimental work and presented a large number of experimental results that may be of interest to readers working in the field of developing information security tools.
The work has a number of disadvantages, the elimination of which will improve the presentation of the material and increase the interest for the reader.

I present these disadvantages below:
1. The title of the work refers to the 5G network, however, it is not clear from the material of the work which particular features of the 5G network are meant. Why are the results obtained focused specifically on promising networks and 5G networks? I agree that useful results were obtained in the work, but from the material of the work I see that they can be used both in modern and promising communication networks. If you keep the mention of 5G in the title, then in the material it is necessary to add explanations of how exactly the proposed method is convenient for 5G networks.

2. Table 1 shows interesting results of the analysis of publications. It is advisable to also provide links to the reviewed sources in this table. This will make it easier to find relevant content.

3. The formulas given in section 3.9 and 3.9.3 are not quite clear. It is necessary to explain the given expressions and decipher the symbols used in them. (If an expression is given, it should be clear to the reader and carry a semantic load).

4. The authors cite a large number of experimental results in section “7. Results". The experiments are described in sufficient detail. I recommend that the authors add a subsection at the end of this section with a brief description of the results obtained, which would outline the essence of the study. I believe that this could be a table that lists the experiments performed and briefly describes the results obtained. This will simplify the perception of the material and highlight the advantages of the proposed method.

I believe that the article can be published after eliminating the indicated shortcomings.

Round 2

Reviewer 1 Report

It is well revised.